# Does Data Scaling Lead to Visual Compositional Generalization?

Arnas Uselis [1]   Andrea Dittadi [2 3 4 5]   Seong Joon Oh [1]

## Abstract

Compositional understanding is crucial for human intelligence, yet it remains unclear whether contemporary vision models exhibit it. The dominant machine learning paradigm is built on the premise that scaling data and model sizes will improve out-of-distribution performance, including compositional generalization. We test this premise through controlled experiments that systematically vary data scale, concept diversity, and combination coverage. We find that compositional generalization is driven by data diversity, not mere data scale. Increased combinatorial coverage forces models to discover a linearly factored representational structure, where concepts decompose into additive components. We prove this structure is key to efficiency, enabling perfect generalization from few observed combinations. Evaluating pretrained models (DINO, CLIP), we find above-random yet imperfect performance, suggesting partial presence of this structure. Our work motivates stronger emphasis on constructing diverse datasets for compositional generalization, and considering the importance of representational structure that enables efficient compositional learning.

## 1. Introduction

Compositional understanding is the ability to comprehend novel, complex scenarios by systematically combining simpler, known conceptual building blocks. It is widely regarded as a cornerstone of human intelligence. The Language of Thought hypothesis suggests that cognition arises from fundamental components and structured recombination

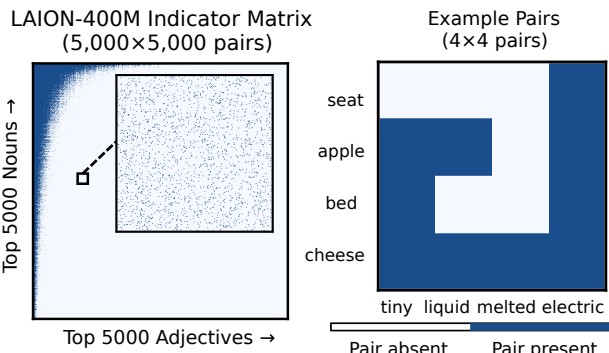

Figure 1: **Sparse concept combinations in large-scale datasets.** Left: An indicator matrix of noun-adjective co-occurrences in LAION-400M shows significant sparsity in concept combinations; the majority of cells are unobserved (zoomed-in view), demonstrating that even common concepts rarely combine in the dataset. This sparsity biases models toward memorizing frequent combinations rather than learning compositional structure. Right: A concrete example of a 4x4 matrix of nouns (seat, apple, bed, cheese) and attributes (tiny, liquid, melted, electric). This work investigates how vision models develop compositional attribute-object understanding in simplified and controlled settings.

rules (Fodor & Fodor, 1975), and neuroscience findings reinforce this perspective (Dehaene et al., 2022). This human proficiency sets a high bar for vision models that must understand how visual attributes and objects combine in novel ways. However, recent studies reveal significant limitations in the compositional abilities of state-of-the-art vision and vision-language models (Rahmanzadehgervi et al., 2024; Tong et al., 2024; Du & Kaelbling, 2024; Yuksekgonul et al., 2023; Zeng et al., 2023), raising fundamental questions about whether and when vision models can achieve this capability.

The dominant paradigm in machine learning relies on scaling data and model size to improve model capabilities, with the expectation that this approach will extend to compositional understanding. This paradigm, grounded in scaling laws (Kaplan et al., 2020; Hoffmann et al., 2022; Hestness et al., 2017) and demonstrated by the success of large language models (Brown et al., 2020; Touvron et al., 2023) and large-scale vision models (Radford et al., 2021; Dosovitskiy et al., 2021), has driven the creation of massive datasets like LAION-400M (Schuhmann et al., 2021). However, as

[1]Tübingen AI Center, University of Tübingen [2]Helmholtz AI [3]Technical University of Munich [4]Munich Center for Machine Learning (MCML) [5]Max Planck Institute for Intelligent Systems, Tübingen. Correspondence to: Arnas Uselis <arnas.uselis@uni-tuebingen.de>.

*Proceedings of the $42^{nd}$ International Conference on Machine Learning*, Vancouver, Canada. PMLR 267, 2025. Copyright 2025 by the author(s).

illustrated in Figure 1, even LAION-400M exhibits critical sparsity in compositional coverage: many plausible attribute-object combinations are rarely or never observed (e.g. "tiny seat" or "melted apple"). This sparsity reflects a combinatorial explosion: with visual attributes (color, shape, texture) that can combine in vast numbers of ways, most possible combinations will remain underrepresented regardless of dataset size.

This motivates our central research question:

*"Do vision models generalize compositionally, and if so, under what conditions?"*

Our approach prioritizes controllability to understand when and how vision models can achieve compositional generalization. We first train models from scratch on carefully designed datasets to isolate the causal effects of data properties on compositional generalization. This allows us to observe both how generalization performance and representational structure emerge under different data conditions. We then validate whether large-scale pretrained vision models exhibit similar structure and examine how this relates to standard linear probing techniques.

Through this controlled approach, we make five contributions:

**(1) Controlled experimental framework (§3)**: We develop a framework (referred to as $(n, k)$-framework) to systematically study how data scaling impacts compositional generalization, varying key factors including training data scale, concept diversity, and combination exposure while focusing on single-object cases to isolate core compositional abilities.

**(2) Data diversity over scale (§4.1)**: We demonstrate that compositional generalization depends critically on data diversity rather than scale: simply increasing in-distribution training data fails to improve generalization, while increasing diversity of data through more concept values and their combinations enhances performance.

**(3) Three-phase feature learning (§4.2)**: We show that models exhibit three phases of feature learning: (i) spurious features with limited diversity, (ii) discriminative but non-linearly-factored features at moderate diversity, and (iii) linearly factored representations only under high diversity.

**(4) Theoretical efficiency of linearly factored structure (§4.3)**: We prove that when representations exhibit linearly factored structure, observing just two combinations per concept value is sufficient for perfect generalization to all unseen combinations.

**(5) Evaluation of pretrained large-scale models (§5)**: We evaluate whether large-scale pretrained models (like DINO and CLIP) exhibit the linearly factored structure identified in our controlled experiments, finding they achieve above-random yet imperfect compositional performance.

Our experiments reveal a clear principle: compositional generalization is driven by data diversity, not mere data scale. Increased combinatorial coverage forces models to discover a linearly factored representational structure, where concepts decompose into additive components. We prove this structure is not just an artifact but a key to efficiency, enabling perfect generalization from just two examples per concept.

## 2. Related Work

**Compositionality, simplicity bias, and generalization.** Compositional understanding—the ability to combine known building blocks into novel representations—is a cornerstone of human intelligence (Fodor & Fodor, 1975; Dehaene et al., 2022). A central question in machine learning is whether neural networks can achieve this systematic generalization. While formalisms for compositionality have been proposed through complexity-based theories (Elmoznino et al., 2024), structural analyses (Lepori et al., 2023), and risk minimization frameworks (Mahajan et al., 2024), models often exhibit a *simplicity bias* (Valle-Pérez et al., 2018; Ren & Sutherland, 2024). They favor simple, spuriously correlated features over more complex, robust ones (Geirhos et al., 2020), a challenge that causal and concept-based representation learning aims to address (Rajendran et al., 2024). This bias is especially pronounced when some concept combinations are underrepresented, or come from a different domain (Jeong et al., 2025). Our work provides a systematic, empirical investigation into the specific data conditions that compel models to overcome this bias and learn a generalizable, compositional latent structure.

**Role of data and scaling.** The structure of training data is known to be critical for generalization (Madan et al., 2021). Prior work has shown that training on compositionally structured data improves performance (Stone et al., 2017), and that augmenting data with diverse primitive combinations is beneficial in NLP (Zhou et al., 2023). The broader trend of scaling has led to emergent abilities in large language models (Brown et al., 2020; Bubeck et al., 2023), including in-context skill composition (He et al., 2024; Arora & Goyal, 2023). However, fundamental limitations remain (Dziri et al., 2023; Zhao et al., 2024; Yu et al., 2023), and performance is often tied to concept frequency in the pretraining data (Udandarao et al., 2024; Wiedemer et al., 2025). We contribute to this debate by isolating *combinatorial diversity* from raw data quantity, especially when models are trained from scratch, showing that the former is the critical driver for visual compositional generalization, whereas simply increasing the latter is insufficient.

**Structured and linearly factored representations.** A

growing body of work finds that large models often exhibit structured representations. Specifically, in large vision-language models, concept embeddings have been observed to sometimes exhibit (to a certain extent) *linearity in representation space*, where a composite concept's representation is the vector sum of its constituents (Trager et al., 2023; Stein et al., 2024; Park et al., 2024; Andreas, 2019). Theoretical work provides formal conditions under which modularity and abstract representations emerge naturally, for instance as a function of input statistics (Dorrell et al., 2024; Whittington et al., 2022) or when networks are trained to perform multiple tasks (Johnston & Fusi, 2017). However, merely learning structured or disentangled representations does not automatically guarantee compositional generalization, and the precise conditions under which a compositional structure yields such generalization remain an active area of theoretical inquiry (Lippl & Stachenfeld, 2024; Montero et al., 2022; 2020; Dittadi et al., 2020), particularly for visual attributes (Zhu et al., 2024). Our work provides further investigation under compositional generalization viewpoint, demonstrating the three-phase *emergence* of this linear structure as a function of data diversity and proving its efficiency for generalization.

**Model-centric approaches and evaluation frameworks.** Many works aim to improve compositionality through model-centric solutions, such as specialized architectures (Zahran et al., 2024; **?**), object-centric models (Locatello et al., 2020; Wiedemer et al., 2023), soft prompting (Nayak et al., 2023), or feature alignment (Wang et al., 2024a), or algorithmic changes (Ren et al., 2023; 2020). These methods are often studied in zero-shot settings (Atzmon et al., 2020; Xian et al., 2020; Isola et al., 2015; Wang et al., 2023). Concurrently, vision-language models face their own compositional challenges, with debates on whether the bottleneck lies in the vision or text encoder (Du & Kaelbling, 2024; Yuksekgonul et al., 2023; Kamath et al., 2023; Vani et al., 2024). In contrast to these model-focused approaches, our work investigates whether compositionality can emerge naturally in standard architectures, isolating the data's structure as the primary variable. This requires careful evaluation, and while benchmarks exist for complex reasoning (Zerroug et al., 2022) or specific setups (Madan et al., 2021; Schott et al., 2022; Mamaghan et al., 2024), our $(n, k)$ framework is designed as a controlled tool. It allows us to make precise, causal claims about the data factors that drive generalization.

## 3. Approach and experimental framework

In this section, we establish a systematic framework for studying compositional generalization in visual discriminative tasks. We begin by formalizing the compositional generalization through a structured mathematical framework that characterizes how visual concepts combine. We then introduce our $(n, k)$ experimental framework that allows us to systematically control the complexity of concept spaces and evaluate models' ability to generalize to novel concept combinations. Finally, we describe our experimental design, covering both training models from scratch and evaluating pre-trained foundation models.

Our approach is motivated by the question of whether scaling data can enable compositional generalization in vision models. To understand the mechanisms behind learning, we also examine whether models develop structured representations in a form of *linear factorization*, as such structure has been observed to an extent in large pretrained vision models (Stein et al., 2024; Trager et al., 2023).

**Data and concept space.** We start by formalizing how we represent visual data in terms of concepts. Formally, we consider a finite set $\mathcal{C} = \mathcal{C}_1 \times \cdots \times \mathcal{C}_c$ of $c$ concepts representing a factored set of concepts, where each $\mathcal{C}_i$ contains possible concept values for a particular concept (like shape or color). Each image $\mathbf{x} \in \mathcal{X}$ is characterized by its position in this concept space through a mapping that assigns it a value from each $\mathcal{C}_i$. For example, an image of a red square could be represented as a point $c = (c_1, c_2, \ldots, c_c) \in \mathcal{C}$ where $c_1 \in \mathcal{C}_1 = \{\text{red, blue, green}\}$ represents color and $c_2 \in \mathcal{C}_2 = \{\text{square, circle, triangle}\}$ represents shape, and other concepts representing other attributes.

**Definition 3.1** (Concepts and Concept Space). A **concept space** $\mathcal{C} = \mathcal{C}_1 \times \cdots \times \mathcal{C}_c$ is a Cartesian product of $c$ sets, where each set $\mathcal{C}_i$ is called a **concept** and contains all possible values for concept $i$. Each element $c_i \in \mathcal{C}_i$ is called a **concept value**, and each element $c \in \mathcal{C}$ represents a unique combination of concept values $(c_1, \ldots, c_c)$ where $c_i \in \mathcal{C}_i$.

Although real-world images typically contain many concepts (e.g. color, shape, size, texture), we simplify our study by focusing on pairs of concepts—for example, how models combine colors with shapes in new ways. Even with this simplified setup, we find that models struggle significantly (see Section 4), suggesting that handling more concepts simultaneously would be even more difficult.

**The $(n, k)$ framework.** To systematically study compositional generalization, we need a way to control the complexity of concept spaces and the diversity of training data. We introduce the $(n, k)$ framework that characterizes concept combination spaces through two key parameters:

$n$ : number of distinct values each concept can take

$k$ : number of training examples per concept value

Given two concepts with $n$ values each, there are $n^2$ possible combinations forming an $n \times n$ grid. We observe only $k$ combinations for each concept value during training, testing generalization on the remaining unseen com-

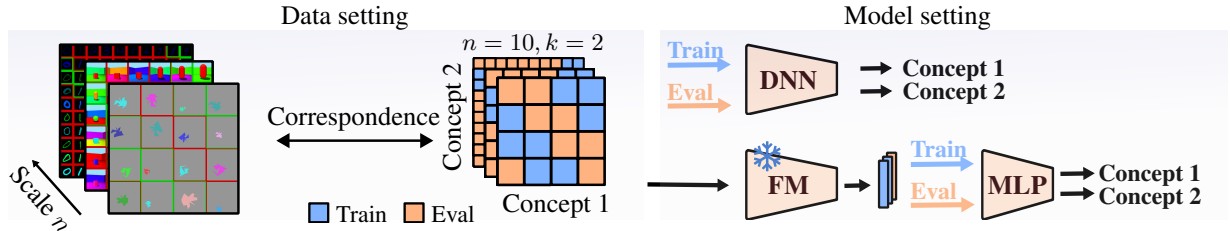

Figure 2: **Investigating compositional learning through concept scaling.** The figure illustrates our two main experimental settings. **Left (Data setting):** Training data consisting of images with corresponding concept combinations shown in the grid, where blue cells indicate observed combinations during training. **Right (Model setting):** Two approaches—training models from scratch (Section 4) where we systematically increase the number of possible concept values $n$ while fixing combinations per concept at $k = 2$, showing examples with $n = 4$, $n = 6$, and $n = 10$, and evaluating pre-trained foundation models' (FM) compositional abilities by fitting an MLP classifier on features (Section 5). The grid demonstrates how the concept space expands as we increase $n$, creating a larger set of unseen combinations for testing generalization.

binations. This framework allows systematic control over both concept complexity ($n$) and training data diversity ($k$).

The figure on the right illustrates training combinations for $n = 4$ concepts with $k = 3$ and $k = 2$ combinations per concept value. Blue cells indicate the set of training combinations, which we denote

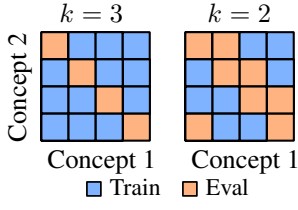

$\mathcal{S}_\text{train}$, while orange cells represent the unseen test combinations, denoted $\mathcal{S}_\text{test}$. Each concept value appears in exactly $k$ training combinations.

**Total dataset size.** For each of the $n \times k$ observed training combinations (the blue cells), we generate multiple images to ensure models learn robustly. Specifically, each image varies along several additional *unlabeled* concept dimensions, $\mathcal{C}_\text{vary}$ (like position, orientation, or background). We sample $n_\text{cell}$ examples for each labeled combination, sampling uniformly across all possible unlabeled variations. For instance, in a setup with $n = 4$ and $k = 2$, there are $n \times k = 8$ distinct labeled training combinations. If we introduce two unlabeled concepts, such as 8 possible positions and 12 possible rotations, the total number of unique images in the training set becomes $8 \times 8 \times 12 = 768$. Concrete examples of the concept space for different values of $n$ and $k$ are shown in Figure 15 in Appendix.

**Compositional generalization.** Having established our concept space framework, we now formalize the specific learning problem we study. Let $\mathcal{X}$ be the space of images and $\{\mathcal{C}_i\}_{i=1}^c$ be the set of possible values for all $c$ concepts. While images vary along all concept dimensions, we focus on learning and evaluating compositional relationships between two consistently labeled concepts. Specifically, each image $\mathbf{x} \in \mathcal{X}$ in our training data is explicitly labeled with a pair of concept values $(c_1, c_2) \in \mathcal{C}_1 \times \mathcal{C}_2$, while all other factors of variation (like position, orientation, or background) remain as unlabeled concepts.

The compositional generalization problem over two concepts can be formalized as follows:

(1) **Training**: Given a dataset $\mathcal{D}_\text{train} = \{(\mathbf{x}_i, c_1^i, c_2^i)\}_{i=1}^{N_\text{train}}$ of $N_\text{train}$ total images, where each image $\mathbf{x}_i$ is explicitly labeled with its concept values (e.g., $c_1$ for color, $c_2$ for shape). The training combinations $(c_1^i, c_2^i)$ are drawn from the restricted subset $\mathcal{S}_\text{train} \subset \mathcal{C}_1 \times \mathcal{C}_2$. We refer to this as *in-distribution (ID)* data.

(2) **Testing**: Evaluate on combinations from $\mathcal{S}_\text{test} = (\mathcal{C}_1 \times \mathcal{C}_2) \setminus \mathcal{S}_\text{train}$, i.e., concept pairs that never co-occurred during training. We refer to this as *out-of-distribution (OOD)* data.

(3) **Goal**: Learn a model $f$ that accurately predicts both labeled concepts $(f_1(\mathbf{x}), f_2(\mathbf{x}))$ even for images containing unseen combinations.

**Experimental design.** Our experimental approach consists of two main settings as illustrated in Figure 2: (1) training models from scratch, and (2) evaluating pretrained foundation models on compositional tasks under our framework. In both cases we systematically vary the $(n, k)$ parameters.

**Representation structure and linearity.** A key question for understanding compositional generalization is how concepts are represented and combined in the learned feature space. We investigate whether concepts combine linearly in the representation space, which would provide a concrete mechanism for efficient compositional generalization (as we show in Section 4.3).

**Definition 3.2** (Linearly factored embeddings (Trager et al., 2023))**.** Given a concept space $\mathcal{C} = \mathcal{C}_1 \times \cdots \times \mathcal{C}_c$, a collection of vectors $\{\mathbf{u}_{c_1}, \ldots, \mathbf{u}_{c_c}\}_{c_1, \ldots, c_c \in \mathcal{C}}$ is linearly factored if there exist vectors $\mathbf{u}_{c_i} \in \mathbb{R}^d$ for all $c_i \in \mathcal{C}_i$ ($i = 1, \ldots, c$), which we refer to as concept representations, such that for all $\mathbf{c} = (c_1, \ldots, c_c)$:

$$\mathbf{u}_c = \mathbf{u}_{c_1} + \cdots + \mathbf{u}_{c_c}. \tag{1}$$

While neural networks are not guaranteed to learn such linearly factored representations, in practice we often observe that these structures emerge during training, as we will

demonstrate in the following sections. When such linear factorizations do emerge, they offer benefits in generalizing compositionally, as we will show in Section 4.3.

**Experimental setup.** The guiding principle for our work was to grant models maximally favorable conditions for demonstrating compositional abilities. We do this through several deliberate choices: using oracle model selection rather than validation, fitting multiple classification heads simultaneously to encourage feature reuse, and partitioning concept combinations to create clear train (ID) / test (OOD) evaluation splits.

**Model selection and metrics.** For model selection, we use the average accuracy across all concepts at each epoch. We perform *oracle* model selection by directly evaluating models on the test set to select the best performing checkpoint (Gulrajani & Lopez-Paz, 2020). This allows us to focus on the fundamental capabilities of models rather than validation strategies.

*(1) Training from scratch:* We use RESNET-50 (He et al., 2015) with linear classification heads; we found that using a transformer backbone (ViT) did not improve generalization performance (see Appendix C.5). The model outputs two predictions $f(\mathbf{x}) = (f_1(\mathbf{x}), f_2(\mathbf{x}))$ where $f_j : \mathcal{X} \to \mathcal{C}_j$ predicts the value of concept $j$ using a shared backbone followed by separate linear heads. Unlike CLIP which uses language embeddings for classification, we learn fixed classification heads directly from visual data to provide an optimistic setting for compositional learning through feature reuse.

*(2) Pre-trained models:* We evaluate RESNET50-IMAGENET1K (He et al., 2015), RESNET50-DINOV1 (Caron et al., 2021), DINOV2-VIT-L/14 (Oquab et al., 2024), and CLIP-VIT-L/14 (Radford et al., 2021). For these models, we pick the best probe architectures on the frozen pre-trained features: a direct linear probe (no hidden layers), an MLP with one hidden layer of size `512`, or an MLP with two hidden layers of size `[512, 512]` with ReLU activations; we found these to provide the best performance, and more complex architectures lead to diminishing returns (results in Appendix C.4).

**Datasets.** We use DSPRITES (Matthey et al., 2017) (using only heart shape to avoid symmetries), 3DSHAPES (Kim & Mnih, 2019), PUG (Bordes et al., 2023), COLORED-MNIST (Arjovsky et al., 2020), and a dataset we introduce of perceptually-challenging shapes without symmetries to which we refer as FSPRITES. Details in Appendix D.

**Metrics.** To evaluate compositional generalization and analyze the learned representations, we use two sets of metrics.

For *generalization*, we report the zero-shot accuracy on $\mathcal{S}_{\text{test}}$, measuring the model's ability to classify unseen concept combinations. We report the average accuracy for the concept pair under consideration.

For representation structure, we consider:

*(i) Decodability*—following Kirichenko et al. (2023); Uselis & Oh (2025), we train linear probes on balanced data and report average accuracy across concepts, indicating if features capture concept information; that is, we merge the training and testing sets, and use a held-out dataset covering all concept combinations for measuring decoded accuracy.

*(ii) Linearity*—we compute the coefficient of determination ($R^2$) between joint representations $\mathbf{f}(\mathbf{x})$ and their reconstruction from individual concept representations $\sum_{i=1}^{k} \mathbf{u}_{c_i}$, where $R^2 = 1 - \frac{\sum_{\mathbf{x}} \|\mathbf{f}(\mathbf{x}) - \sum_{i=1}^{k} \mathbf{u}_{c_i}\|^2}{\sum_{\mathbf{x}} \|\mathbf{f}(\mathbf{x}) - \bar{\mathbf{f}}\|^2}$ with $\bar{\mathbf{f}} = \frac{1}{|\mathcal{D}|} \sum_{\mathbf{x} \in \mathcal{D}} \mathbf{f}(\mathbf{x})$ measures how well representations follow linear structure. Here, $\bar{\mathbf{f}}$ represents the mean representation across all samples.

*(iii) Orthogonality*—we measure the mean cosine similarity $\frac{1}{|\mathcal{C}_1||\mathcal{C}_2|} \sum_{i,j} \cos(\mathbf{u}_{c_1^i}, \mathbf{u}_{c_2^j})$ between concept representations to assess if concepts are encoded in orthogonal subspaces, sometimes found in pretrained models (Stein et al., 2024; Wang et al., 2024b).

We report the representation structure metrics only for from-scratch models; this is due to the fact that pretrained models may encode other information other than the target concepts.

## 4. Does compositional generalization emerge with data scale?

Building on our formal framework, we systematically investigate how neural networks learn compositional understanding as we vary both data quantity and concept diversity. Our $(n, k)$ framework allows us to precisely control which concept combinations models see during training, enabling us to isolate how different factors affect compositional generalization. Through controlled experiments, we investigate several key questions:

1. Can models generalize compositionally under basic settings? We find that compositional generalization remains challenging with accuracy drops of 27-95% on unseen combinations.
2. Does increasing ID data quantity improve compositional generalization? We show that simply scaling ID data quantity is insufficient.
3. Can neural networks achieve compositional generalization under any conditions? Yes, but only with sufficiently diverse training data.
4. What kind of structure do representations exhibit when models generalize well? We find that models that generalize well exhibit a highly linear and orthogonal structure in their feature space.

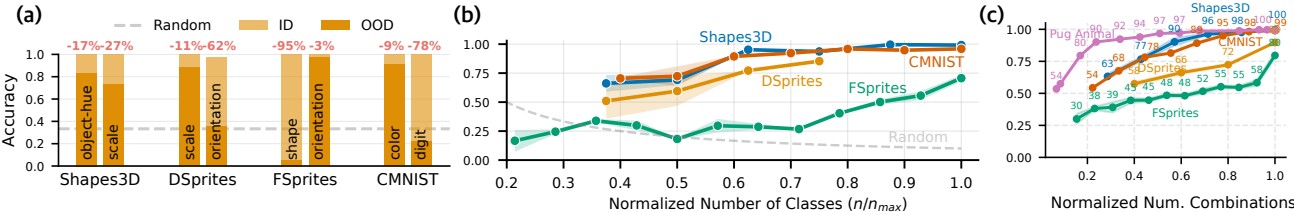

Figure 3: **Compositional generalization emerges through different forms of concept diversity.** (a) In basic settings with limited diversity, models show substantial accuracy drops on unseen combinations (brown) compared to seen combinations (yellow), demonstrating the inherent difficulty of compositional generalization. (b) When increasing the number of target classes ($n$) while keeping dataset size and diagonal training combinations fixed ($k = n - 1$), models show improved generalization, suggesting that target space diversity drives compositional learning. (c) With fixed maximum target classes, increasing the number of training combinations ($k$) also improves performance, showing that exposure to more concept combinations enhances generalization ability, even if the target size is the same.

5. What are the theoretical benefits of such structure for compositional generalization? We show that this linear structure enables perfect generalization to unseen combinations with just two combinations per concept value.

### 4.1. Compositional generalization is difficult but diverse data helps

**Models struggle with basic compositional generalization.** In Figure 3(a), in a basic compositional setting with $n = 3$ concept values and $k = 2$ seen combinations per concept value, while all models achieve strong ID accuracy (near 100%, yellow bars), their performance drops significantly when evaluated on unseen combinations of concepts (brown bars). For example, MNIST digit recognition accuracy drops by around 78% in the OOD setting. Interestingly, in all datasets, at least one concept shows relatively small degradation, ranging from only 3% drop (orientation in FSprites) to 17% drop (object-hue in Shapes3D), while other concepts in the same datasets show much larger performance gaps.

**Increasing concept diversity improves generalization.** Figure 3(b,c) shows that generalization improves both when increasing the number of target classes ($n$) with fixed diagonal training combinations ($k = n - 1$), and when increasing training combinations ($k$) with fixed maximum target classes. This suggests that both target space diversity and exposure to more concept combinations enhance compositional learning, even when the target size remains constant.

**Dataset size alone provides limited improvement for generalization.** We experimented with RESNET50 trained from scratch using $n = 3, k = 1$ and three different training set sizes: 7,500, 15,000, and 30,000 samples for SHAPES3D and CMNIST (the maximum number of unique samples possible with these combinations), and up to 120,000 samples for DSPRITES and FSPRITES. We excluded PUG from this analysis since with $n = 3$, there were too few unique samples available to effectively train the model from scratch.

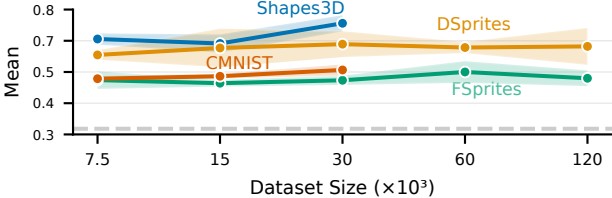

Figure 4: **Increasing ID training data quantity does not solve compositional generalization.** Despite training with significantly more in-distribution samples, models still struggle to generalize to unseen concept combinations. The gap between ID and OOD performance remains large across all datasets, suggesting that the challenge of compositional generalization cannot be solved simply by scaling up training data within the same distribution.

As shown in Figure 4, despite increasing the training data by 4x, the gap between ID and OOD performance remains large across all datasets: models still show accuracy drops of 60-80% on unseen combinations. This suggests that simply scaling up training data within the same distribution is insufficient for achieving compositional generalization.

> **Takeaway §4.1:** Compositional generalization remains challenging across all datasets, with accuracy drops of 60-80% on unseen combinations despite perfect in-distribution performance. While increasing target diversity and combination exposure improves generalization, scaling dataset size provides limited improvement. Some concepts show relatively small degradation (3-17% drops) while others in the same datasets show much larger gaps. Both target space diversity and exposure to more concept combinations enhance compositional learning, but increasing training data quantity (up to 4x) only helps reduce the large ID-OOD performance gap without fully solving the problem.

### 4.2. Three-phase behavior in feature learning

To understand why models struggle with compositional generalization, we investigate two potential explanations motivated by prior work on shortcut learning and distributional robustness (Geirhos et al., 2020; Sagawa et al., 2020). First, the learned features could be spurious, failing to capture meaningful concept information. Second, novel concept

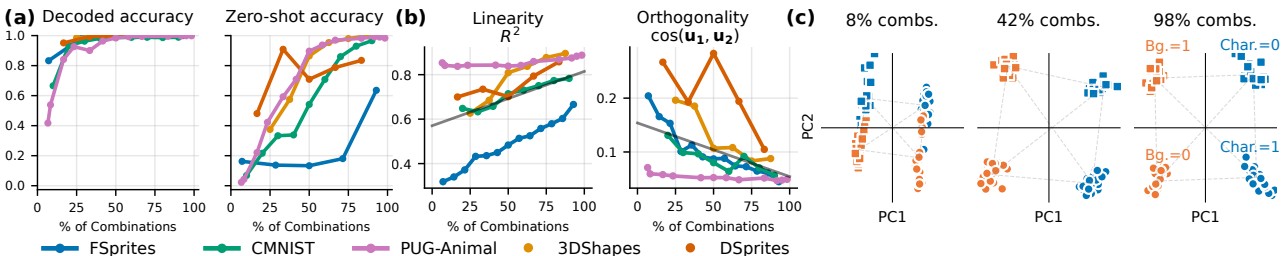

Figure 5: **Linearity emerges with data diversity, while feature discriminability alone does not imply linear structure.** (a) Feature discriminability emerges early but does not imply compositional structure, (b) Linear concept representations only emerge with increased training diversity, as shown through $R^2$ scores and orthogonality measures, (c) PCA visualizations confirm evolution from entangled to linear feature organization as training diversity increases. X-axis represents percentage of training combinations $k/n$, with $n$ being the maximum number of concept values.

combinations may produce "misplaced" representations that the classifier fails on. We analyze these possibilities by examining the structure of feature spaces using the linearity and orthogonality metrics defined in Section 3, measuring both the quality of individual concept representations and how predictably they combine. Using a balanced dataset with all concept combinations (including unseen ones) and 100 samples per combination, we evaluate models trained in the previous section across multiple datasets (MNIST, FSPRITES, SHAPES3D, PUG).

Our analysis reveals two key findings about how neural networks learn to represent concepts (Figure 5). First, we find that *linearity in representations* emerges naturally as models are exposed to more diverse training combinations. As shown in Figure 5(b), both the linear separability ($R^2$ scores) and orthogonality (cosine similarity) of concept dimensions improve with increased training diversity. This emergence of linear structure is accompanied by improved zero-shot generalization—Figure 5(a) shows that zero-shot accuracy on unseen combinations steadily increases as training diversity grows.

Second, we observe that this progression occurs in three distinct phases: (i) With limited concept combinations (0-10%), models learn spurious features with poor discrimination (decoded accuracy <80%) and random-level zero-shot performance, as shown by entangled representations in Figure 5(c) at 8%.

(ii) At moderate diversity (25-75%), linearity and orthogonality begin emerging (Figure 5(b)), with features becoming decodable (100% accuracy) and zero-shot performance reaching 60-80%.

(iii) At high diversity (75-100%), while discriminability plateaus, representations become strongly linear ($R^2 > 0.8$) and orthogonal (cosine similarity <0.1), enabling zero-shot accuracy above 90% on the majority of the datasets. The PCA visualizations in Figure 5(c) qualitatively confirm this progression from entangled to linear factorization.

These results indicate a link between training diversity and representation structure in NNs. While models can learn to discriminate individual concepts with limited data (at around 25%), linearity in representations emerges only with extensive concept diversity. Empirically, linearity and zero-shot accuracy appear to be directly related, suggesting an explanation of previous work showing that decodable features can be re-aligned to support generalization in large systems like CLIP (Koishigarina et al., 2025).

> **Takeaway §4.2:** Neural networks exhibit three phases: (1) With limited diversity (<10% combinations), models learn spurious features and fail at basic concept discrimination; (2) At moderate diversity (10-75% combinations), models gain discriminative ability but lack linear structure; (3) Only with high diversity (>75% of combinations) does true compositional structure emerge, with highly linear ($R^2 > 0.8$) and orthogonal (cosine similarity < 0.2) concept dimensions. This progression shows that concept diversity is necessary for models to learn structured and generalizable representations.

### 4.3. Benefits of linear factorization

The benefit of a linear feature structure becomes apparent when contrasted with the weaker property of decodability. While features are often *decodable*, this alone is insufficient for generalization to unseen combinations. Generalizing through decodability may require exposure to all possible concept pairings, which is infeasible. As illustrated in Figure 7 (center), while adaptation can compensate for unstructured representations, this approach demands a balanced dataset of all combinations, which is impractical at scale. In contrast, a *linear* feature structure enables generalization without exhaustive supervision. As shown in Figure 7 (right), when representations are organized linearly, models can correctly classify novel combinations, overcoming the limitations of mere decodability.

Motivated by our observation that models achieving strong compositional generalization exhibit highly linear concept representations, we now investigate the theoretical benefits of such a structure. In this idealized case, how many

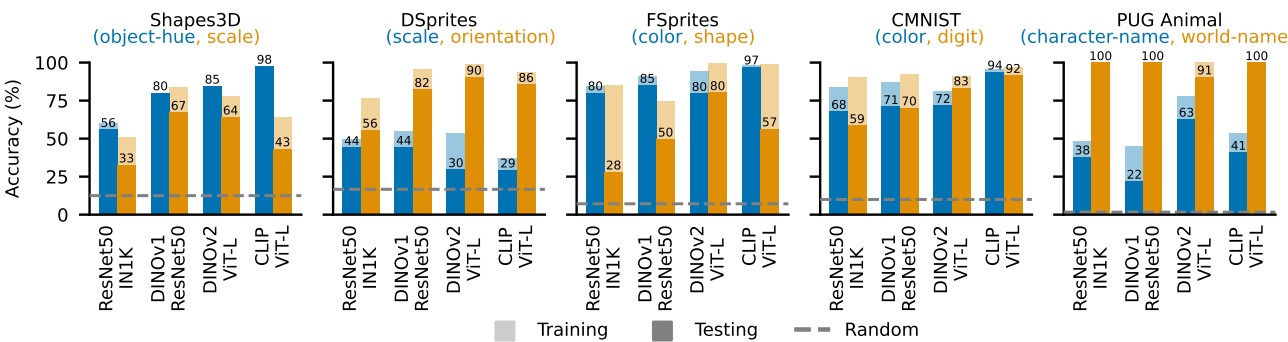

Figure 6: **Compositional generalization capabilities of pre-trained models under assumed linear factorization.** Bar plots show both training (transparent) and testing (solid) accuracy across different datasets (DSPRITES, SHAPES3D, CMNIST, PUG-ANIMAL) when using minimal training data ($k = 2$ combinations per concept) to learn linear concept representations for each concept. Dashed lines indicate random baseline performance. Following Proposition 4.1, we identified the factored representations $\mathbf{u}_{c_1}$ and $\mathbf{u}_{c_2}$ for each concept value using $k = 2$ combinations per concept value. While perfect generalization predicted by the proposition would require ideal linear compositionality, our empirical results show strong performance on certain concepts (e.g., > 90% accuracy on color, orientation, digit, and background concepts for either CLIP or DINOv2 models), with varying effectiveness across different concept types and models, suggesting that pre-trained representations exhibit partial linearity in their representations.

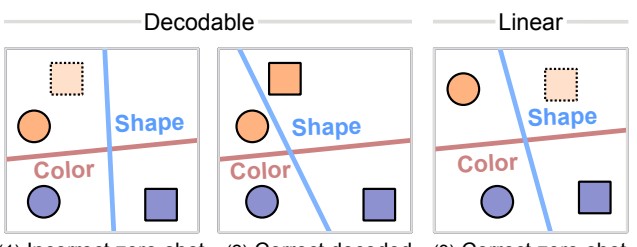

Figure 7: **Importance of linear feature structure for compositional generalization.** We illustrate a schematic for shape and color classification using linear models in a 2-dimensional feature space, comparing zero-shot and adapted cases with frozen feature extractor. (1) If the feature space lacks a **linear structure**, the model misclassifies the `orange square` in zero-shot inference. (2) Adaptation by adding `orange square` samples allows correct classification. (3) A **linearly structured** feature space enables correct zero-shot generalization without adaptation. The decision boundaries are linear in all cases, but only the features in the rightmost panel enable zero-shot generalization.

concept combinations would a model with perfectly linear representations need to observe to generalize to all unseen combinations? We answer this questions in the following proposition.

**Proposition 4.1** (Minimal Compositional Learning). *Let $f : \mathcal{X} \to \mathbb{R}^d$ be a feature extractor with linearly factored concept embeddings over $\mathcal{C}$. Let $\{\mathbf{u}_{c_1^1}, \ldots, \mathbf{u}_{c_1^n}\}$ and $\{\mathbf{u}_{c_2^1}, \ldots, \mathbf{u}_{c_2^n}\}$ be the concept vectors for the first and second concepts respectively, where their joint span has dimension $2n - 1$. Suppose we only observe joint representations for concept combinations $c_i, c_j \in \{1, \ldots, n\}$. Then $k = 2$ combinations per concept value suffice to learn a linear classifier that perfectly generalizes to all $(n - k) \cdot n$ unseen combinations.*

This proposition illustrates the benefit of perfectly composi-

tional representations: with just two examples per concept value, perfect generalization is possible if the feature space is linearly factorized. We view this as a starting point—while the assumption of linearly independent factors is often satisfied in both from-scratch and pre-trained models, it can break down as the number of values grows, making joint linear independence impossible - e.g., such factors may occupy low-dimensional subspaces (Sonthalia et al., 2025). We expect that this assumption can be relaxed, and that a more complete understanding of the setting is possible in future work.

> **Takeaway §4.3:** When linear factorization is present, perfect compositional generalization is possible with just two combinations per concept value.

## 5. Do large pre-trained models generalize compositionally?

Our analysis of models trained from scratch revealed that linear structure emerges naturally when models are exposed to diverse concept combinations. This finding raises a question: Have large-scale pretrained models already learned such linear structure through their pretraining? To investigate this, we evaluate pretrained models using two complementary approaches. We first test for the ideal linear structure from our theoretical framework (Proposition 4.1), which would enable perfect generalization. This reveals how close existing models are to this optimal linear structure. Second, we use (non-)linear probing to assess general concept accessibility in the feature space. Comparing these approaches allows us to distinguish between models that simply encode concept information and those that represent it in a structured, linear manner.

## 5.1. Evaluating via linear factorization

**Measuring linearity.** Building on our earlier findings showing the natural emergence of linearly factored representations, we test how well the recovered concept value representations (detailed algorithm in Appendix 1) can be used to classify novel concept combinations. Classification of a new input $\mathbf{x}$ can then be performed by projecting the representation $f(\mathbf{x})$ onto the $\mathbf{u}$ and $\mathbf{v}$ values to acquire labels for both concepts.

We calculate accuracy for each concept using this approach and illustrate the results in Figure 6. Certain concept pairs show strong amenability to linear representation across all models. On PUG-ANIMAL, all models achieve exceptionally high accuracy (>90%) on WORLD-NAME concept, suggesting more linear representations. The best model consistently exceeds 90% accuracy on *some* concept classification across all datasets. Additionally, models show clear specialization: CLIP excels at color-based tasks (highest accuracy on CMNIST color-digit and SHAPES3D object-hue), while DINOV2 performs best on shape-based concepts (e.g. on scale, shape, orientation, and character).

While no model achieves the perfect generalization predicted by our theoretical analysis for ideally linear representations, these results demonstrate that pre-trained models exhibit partial linearity in their representations, varying in strength across concept types. Strong performance on some concept pairs supports our hypothesis that linear representation organization facilitates compositional generalization.

## 5.2. Evaluating generalization via probing

While the linear factorization analysis tests for an ideal compositional structure, we also employ a more direct test of generalization: probing. In this approach, we train a simple classifier (a non-linear probe) on the model's features for the *seen* concept combinations from our training set and evaluate it on the *unseen* combinations. This directly measures whether a consistent mapping from features to concepts can be learned and transferred. For each model and dataset, we compute the average accuracy for a given $k$ value, keeping $n = n_{\max}$. To enable fair comparison across datasets, we normalize each model's performance by its maximum accuracy and aggregate the results, as shown in Figure 8.

All pre-trained models consistently outperform the from-scratch RESNET50, showing that pre-training provides a significant advantage. However, it is not a complete solution, as all models improve as the diversity of training combinations increases. Full results are in Appendix C.2.

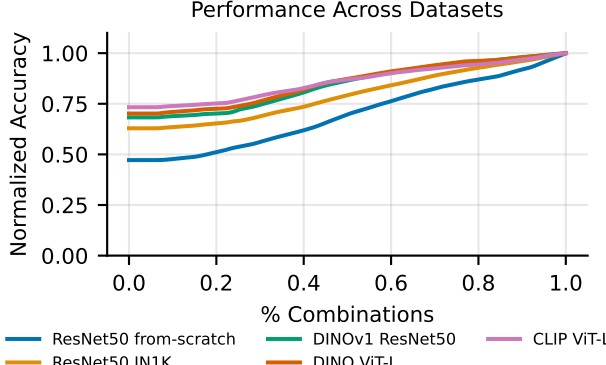

Figure 8: **Even with pretraining, models struggle with compositional generalization.** Despite the benefits of pretraining, models still face challenges in generalizing to unseen concept combinations. While larger models like CLIP and DINO VIT-L show the strongest performance, the persistent gap between pretrained and from-scratch models indicates that current pretraining approaches do not generalize compositionally well.

> **Takeaway §5:** Pre-training is not a substitute for data diversity. While large models like CLIP and DINO VIT-L develop partially linear representations, our analysis shows they only generalize reliably after training a downstream model on a diverse set of concept combinations.

# 6. Conclusion

In this work, we systematically investigated the conditions under which vision models achieve compositional generalization, focusing on the distinct roles of data scale versus data diversity. Our findings reveal that merely increasing the volume of training data is insufficient for generalization to novel concept combinations. Instead, data diversity is the critical factor. We identified a three-phase learning dynamics where models transition from learning spurious correlations to discriminative features, and finally to a linearly structured representation space only when trained with sufficient combinatorial diversity. We provide theoretical evidence for the power of this structure, proving that such linear factorization allows for perfect generalization from a minimal number of training examples in an idealized setting. When we evaluated large-scale pretrained models through this lens, we found they exhibit some of this compositional structure but remain far from perfect, achieving mixed results that highlight their limitations.

Ultimately, our work suggests that while current scaling paradigms are beneficial, they do not automatically confer robust compositional abilities due to the inherent combinatorial sparsity of large-scale datasets. Achieving compositional generalization will likely require a more deliberate focus on structured data diversity to induce the necessary representational geometry in vision models.

## Acknowledgments

We thank the anonymous reviewers for their valuable feedback, Yujin Jeong, Simon Buchholz, Yi Ren, Samuel Lippl, Ankit Sonthalia, Alexander Rubinstein, and Martin Gubri for helpful discussions, and the International Max Planck Research School for Intelligent Systems (IMPRS-IS) for supporting Arnas Uselis. This work was supported by the Tübingen AI Center.

## Impact statement

This work advances understanding of compositional learning in vision models, which could enable more data-efficient and reliable AI systems. We release our code and datasets publicly to promote reproducible research and responsible development of these capabilities.

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

# A. Appendix

# A. Experimental setup and implementation

### A.1. Implementation details

In this section we provide additional details on the implementation of the experiments.

**Optimization.** All models are trained using the Adam (Kingma & Ba, 2017) optimizer. Based on an initial grid search, we use a learning rate of $10^{-4}$ for ResNet training from scratch and $10^{-3}$ for probing pre-trained features. All models are trained for 100 epochs with a batch size of 64.

**Train/Test splits.** For each concept value $i$, we observe combinations with values $j$ where $(i - j + n) \bmod n < k$, and evaluate on all other combinations. This creates a clear distinction between combinations seen during training and those requiring compositional generalization.

The key idea is creating a training set that is balanced such that each concept value is observed with equal frequency. For each concept value $i \in \{0, \ldots, n-1\}$, we observe exactly $k$ combinations during training, defining our training and test sets as:

$$
\begin{aligned}
\mathcal{C}_{\text{train}} &:= \bigcup_{i=1}^{n} \{(i, (i+j \bmod n)) : j \in \{0, \ldots, k-1\}\}, \\
\mathcal{C}_{\text{test}} &:= (\mathcal{C}_1 \times \mathcal{C}_2) \setminus \mathcal{C}_{\text{train}}.
\end{aligned}
\tag{2}
$$

This construction ensures that: (1) each concept value appears in exactly $k$ training combinations, (2) the test set contains $(n - k) \cdot n$ novel combinations, and (3) the split is deterministic and reproducible across experiments.

**Concept value selection.** For each experiment with parameters $n$ and $k$, we select $n$ values for each of our two target concepts that are maximally spread across their respective concept spaces. Specifically, if a concept has $|\mathcal{C}_{\text{max}}|$ possible values, we select values at indices $\{i \cdot \lfloor |\mathcal{C}_{\text{max}}|/n \rfloor\}_{i=0}^{n-1}$ to ensure even coverage.

**Sampling procedure.** Within each valid training combination (each "cell" in our concept grid), we sample $n_{\text{cell}}$ examples uniformly from all possible variations of the remaining unlabeled concepts $\mathcal{C}_{\text{vary}}$ (like position, orientation, background, etc.). This uniform sampling across $|\mathcal{C}_{\text{vary}}|$ possible variations ensures balanced representation of each concept combination across different visual contexts.

# B. Proofs

In this section we provide the proofs for our main theoretical results.

**Notation.** We summarize the notation used throughout the proofs, though we reintroduce each term where appropriate.

- *Spaces and mappings:*
    - $\mathcal{X}$ represents the input space (images)
    - $\mathcal{C} = \mathcal{C}_1 \times \mathcal{C}_2 \times \cdots \times \mathcal{C}_c$ represents the concept space
    - $\mathcal{C}_i$ is the $i$-th concept dimension (e.g., color, shape)
    - $c : \mathcal{X} \to \mathcal{C}$ is the mapping from images to concept values
    - $c(\mathbf{x}) = (c_1, \ldots, c_c)$ gives the concept values for image $\mathbf{x}$
    - $c_i$ denotes the value of the $i$-th concept
- *Framework parameters:*
    - $n$ is the number of concept values per dimension in the $(n, k)$ framework
    - $k$ is the number of training combinations per concept value
    - $c$ is the total number of concept dimensions
- *Feature representations:*
    - $f(\mathbf{x})$ is the feature extractor output for image $\mathbf{x}$
    - $\mathbf{f} = \frac{1}{|\mathcal{D}|} \sum_{\mathbf{x} \in \mathcal{D}} f(\mathbf{x})$ is the global mean embedding
    - $\mathbf{u}_{c_i}$ represents the true concept vector for value $c_i$
    - $\mathbf{u}'_{c_i}$ is the recovered centred concept vector for value $c_i$

- $\mathbf{u}'_{c_i, c_j}$ denotes the pairwise joint embedding for values $c_i, c_j$
- *Datasets:*
    - $\mathcal{D}$ represents a dataset of image-concept pairs
    - $\mathcal{D}_{\mathcal{C}}$ is the dataset over all possible concept combinations
    - $\mathcal{D}_{\text{train}}$ and $\mathcal{D}_{\text{test}}$ are the training and test datasets with limited and unseen combinations, respectively
    - $\mathcal{D}_{c_i}$ is the subset of $\mathcal{D}$ containing concept value $c_i$
    - $\mathcal{D}_{c_i, c_j}$ contains both values $c_i$ and $c_j$
- *Training constructs:*
    - $\mathcal{C}_{\text{train}}$ is the set of observed concept combinations during training
    - $\bar{\mathbf{f}}_{i,j}$ represents the mean embedding for combination $(i, j)$

Let $\mathcal{X}$ denote the input space and $\mathcal{C} = \mathcal{C}_1 \times \mathcal{C}_2 \times \cdots \times \mathcal{C}_c$ represent the concept space. We assume a mapping $c : \mathcal{X} \to \mathcal{C}$ that identifies for each image $\mathbf{x} \in \mathcal{X}$ its corresponding concept values $c(\mathbf{x}) = (c_1, \ldots, c_c) \in \mathcal{C}$.

We denote $\mathcal{D}_{\mathcal{C}}$ as the dataset over all possible concept combinations. In practise, we only observe limited combinations, as discussed in Section A. We denote such a dataset as $\mathcal{D}_{\text{train}}$ and $\mathcal{D}_{\text{test}}$ for the training and test sets, respectively.

We also restate the linear factorization definition from the main text:

**Definition B.1** (Linearly factored embeddings (Trager et al., 2023)). Given a concept space $\mathcal{C} = \mathcal{C}_1 \times \cdots \times \mathcal{C}_c$, a collection of vectors $\{\mathbf{u}_c\}_{c \in \mathcal{C}}$ is linearly factored if there exist vectors $\mathbf{u}_{c_i} \in \mathbb{R}^d$ for all $c_i \in \mathcal{C}_i$ $(i = 1, \ldots, c)$, which we refer to as concept representations, such that for all $\mathbf{c} = (c_1, \ldots, c_c)$:

$$\mathbf{u}_c = \mathbf{u}_{c_1} + \cdots + \mathbf{u}_{c_c}. \tag{3}$$

Assuming linear factorization,

$$f(\mathbf{x}) = \sum_{\ell=1}^{k} \mathbf{u}_{c_\ell(\mathbf{x})},$$

and given a dataset $\mathcal{D} = \{(\mathbf{x}_j, \mathbf{c}_j)\}_{j=1}^{s}$ with $s := \prod_{i=1}^{c} |\mathcal{C}_i|$, with image–concept pairs, we can recover a representation (up to a global shift shared by all factors) for each concept value by averaging feature vectors across all combinations that contain that value (Trager et al., 2023). Formally, for a value $c_i \in \mathcal{C}_i$ let

$$\mathbf{u}'_{c_i} := \frac{1}{|\mathcal{D}_{c_i}|} \sum_{\mathbf{x} \in \mathcal{D}_{c_i}} [f(\mathbf{x}) - \mathbf{f}], \qquad \mathbf{f} := \frac{1}{|\mathcal{D}|} \sum_{\mathbf{x} \in \mathcal{D}} f(\mathbf{x}), \tag{4}$$

Thus $\mathbf{u}'_{c_i}$ is the conditional mean feature vector, centred by the global mean $\mathbf{f}$.

We first describe the relationship between the ground truth factors $\mathbf{u}_{c_i}$ and the recovered ones $\mathbf{u}'_{c_i}$. These relationships only hold for the case when the contstructed factors are recovered from the full dataset.

**Lemma B.2** (Relation to ground truth concept vectors). *Let $\mathbf{u}_{c_i}$ denote the true concept vector for value $c_i$, and $\mathbf{u}'_{c_i}$ the recovered one from* (4). *Over the full dataset,*

$$\mathbf{u}'_{c_i} = \mathbf{u}_{c_i} - \frac{1}{|\mathcal{C}_i|} \sum_{c'_i \in \mathcal{C}_i} \mathbf{u}_{c'_i}.$$

*Proof.* Start from the definition (4) and substitute the linear factorisation $f(\mathbf{x}) = \sum_{\ell=1}^{c} \mathbf{u}_{c_\ell(\mathbf{x})}$:

$$\mathbf{u}'_{c_i} = \frac{1}{|\mathcal{D}_{c_i}|} \sum_{\mathbf{x} \in \mathcal{D}_{c_i}} [f(\mathbf{x}) - \mathbf{f}]$$

$$= \frac{1}{|\mathcal{D}_{c_i}|} \sum_{\mathbf{x} \in \mathcal{D}_{c_i}} \sum_{\ell=1}^{c} \mathbf{u}_{c_\ell(\mathbf{x})} - \mathbf{f}. \tag{1}$$

Interchange the sums in (1). For the term with $\ell = i$ each $\mathbf{x} \in \mathcal{D}_{c_i}$ contributes $\mathbf{u}_{c_i}$, hence

$$\frac{1}{|\mathcal{D}_{c_i}|} \sum_{\mathbf{x} \in \mathcal{D}_{c_i}} \mathbf{u}_{c_i} = \mathbf{u}_{c_i}.$$

For any $\ell \neq i$ each value $c'_\ell \in \mathcal{C}_\ell$ occurs equally often inside $\mathcal{D}_{c_i}$, namely $|\mathcal{D}_{c_i}|/|\mathcal{C}_\ell|$ times. Therefore

$$\frac{1}{|\mathcal{D}_{c_i}|} \sum_{\mathbf{x} \in \mathcal{D}_{c_i}} \mathbf{u}_{c_\ell(\mathbf{x})} = \frac{1}{|\mathcal{C}_\ell|} \sum_{c'_\ell \in \mathcal{C}_\ell} \mathbf{u}_{c'_\ell}.$$

Summing these contributions and using the explicit formula for the global mean

$$\mathbf{f} = \frac{1}{|\mathcal{D}|} \sum_{\mathbf{x} \in \mathcal{D}} f(\mathbf{x}) = \sum_{\ell=1}^{c} \frac{1}{|\mathcal{C}_\ell|} \sum_{c'_\ell \in \mathcal{C}_\ell} \mathbf{u}_{c'_\ell},$$

it follows that

$$\mathbf{u}'_{c_i} = \mathbf{u}_{c_i} + \sum_{\ell \neq i} \frac{1}{|\mathcal{C}_\ell|} \sum_{c'_\ell} \mathbf{u}_{c'_\ell} - \mathbf{f} = \mathbf{u}_{c_i} - \frac{1}{|\mathcal{C}_i|} \sum_{c'_i \in \mathcal{C}_i} \mathbf{u}_{c'_i},$$

as claimed. $\qquad \square$

It also follows that this construction of factors $\mathbf{u}_{c_i}$ leads to recovery of the sum of factored embeddings up to a global mean. Importantly, if full dataset $\mathcal{D}_\mathcal{C}$ is available, normalizing the mean of the embeddings (i.e. setting $\mathbf{f} := \mathbf{0}$) is possible.

**Lemma B.3** (Reconstruction of a centred embedding). *For any* $\mathbf{x}$ *with concept values* $(c_1(\mathbf{x}), \dots, c_c(\mathbf{x}))$

$$f(\mathbf{x}) = \mathbf{f} + \sum_{i} \mathbf{u}'_{c_i(\mathbf{x})}.$$

*Proof.* Using Lemma B.2 we have for every concept value $c_i$

$$\mathbf{u}'_{c_i} = \mathbf{u}_{c_i} - \frac{1}{|\mathcal{C}_i|} \sum_{c'_i \in \mathcal{C}_i} \mathbf{u}_{c'_i}.$$

Applying this identity to the particular values $c_i(\mathbf{x})$ of the sample $\mathbf{x}$ and summing over $i = 1, \dots, k$ yields

$$\sum_{i=1}^{c} \mathbf{u}'_{c_i(\mathbf{x})} = \sum_{i=1}^{c} \mathbf{u}_{c_i(\mathbf{x})} - \sum_{i=1}^{c} \frac{1}{|\mathcal{C}_i|} \sum_{c'_i \in \mathcal{C}_i} \mathbf{u}_{c'_i} = f(\mathbf{x}) - \mathbf{f},$$

where the last equality uses $f(\mathbf{x}) = \sum_i \mathbf{u}_{c_i(\mathbf{x})}$ and the definition of the global mean $\mathbf{f}$. $\qquad \square$

In what follows we study compositional settings where the concept space may include many factors, but only two factors, $\mathcal{C}_1$ and $\mathcal{C}_2$, are observed; the remaining factors $\mathcal{C}_3, \dots, \mathcal{C}_c$ are unobserved. Importantly, factors $\mathcal{C}_1$ and $\mathcal{C}_2$ exhibit a correlation due to the $(n, k)$ framework.

Next, we establish a convenient property of the factored representations.

**Lemma B.4** (Zero-sum embeddings). *For any concept dimension* $i \in \{1, \dots, c\}$,

$$\sum_{c_i \in \mathcal{C}_i} \mathbf{u}'_{c_i} = \mathbf{0}.$$

*Proof.* Let $\mathbf{f} := \frac{1}{|\mathcal{D}|} \sum_{\mathbf{x} \in \mathcal{D}} f(\mathbf{x})$ be the global mean. For each value $c_i \in \mathcal{C}_i$ set

$$\mathcal{D}_{c_i} := \{\mathbf{x} \in \mathcal{D} \mid c_i(\mathbf{x}) = c_i\}, \qquad m := |\mathcal{D}_{c_i}| \ \ (\text{same for every } c_i),$$

Summing over $c_i$ gives

$$\sum_{c_i \in \mathcal{C}_i} \mathbf{u}'_{c_i} = \sum_{c_i \in \mathcal{C}_i} \left[ \tfrac{1}{m} \sum_{\mathbf{x} \in \mathcal{D}_{c_i}} f(\mathbf{x}) - \mathbf{f} \right] \tag{5}$$

$$= \tfrac{1}{m} \sum_{c_i \in \mathcal{C}_i} \sum_{\mathbf{x} \in \mathcal{D}_{c_i}} f(\mathbf{x}) - |\mathcal{C}_i| \mathbf{f} \tag{6}$$

$$= \tfrac{1}{m} \sum_{\mathbf{x} \in \mathcal{D}} f(\mathbf{x}) - |\mathcal{C}_i| \mathbf{f} \tag{7}$$

$$= \tfrac{|\mathcal{D}|}{m} \mathbf{f} - |\mathcal{C}_i| \mathbf{f} \quad (|\mathcal{D}| = |\mathcal{C}_i| m) \tag{8}$$

$$= |\mathcal{C}_i| \mathbf{f} - |\mathcal{C}_i| \mathbf{f} = \mathbf{0}. \tag{9}$$

$\square$

In practice, we often only observe a subset of concept combinations. To accomodate such a constraint, we formalize it through pairwise joint embeddings:

**Definition B.5** (Pairwise joint embedding). Given a concept space $\mathcal{C} = \mathcal{C}_1 \times \cdots \times \mathcal{C}_c$, the pairwise joint embedding for factors $i \neq j$ and values $c_i \in \mathcal{C}_i, \; c_j \in \mathcal{C}_j$ is

$$\mathbf{u}'_{c_i,c_j} = \frac{1}{|\mathcal{D}_{c_i,c_j}|} \sum_{\mathbf{x} \in \mathcal{D}_{c_i,c_j}} \left[ f(\mathbf{x}) - \mathbf{f} \right], \qquad \mathcal{D}_{c_i,c_j} := \{ \mathbf{x} \in \mathcal{D} \mid c(\mathbf{x})_i = c_i, \; c(\mathbf{x})_j = c_j \}. \tag{10}$$

**Lemma B.6** (Additivity of joint embeddings). *Under a linear factorisation s.t.* $f(\mathbf{x}) = \sum_{\ell=1}^c \mathbf{u}_{c_\ell(\mathbf{x})}$ *holds,*

$$\mathbf{u}'_{c_i,c_j} = \mathbf{u}'_{c_i} + \mathbf{u}'_{c_j}. \tag{11}$$

*Proof.* Define

$$\mathcal{D}_{c_i,c_j} := \{ \mathbf{x} \in \mathcal{D} \mid c(\mathbf{x})_i = c_i, \; c(\mathbf{x})_j = c_j \}, \qquad N_{c_i,c_j} := |\mathcal{D}_{c_i,c_j}|.$$

Substituting the centred decomposition $f(\mathbf{x}) = \mathbf{f} + \sum_{\ell=1}^c \mathbf{u}'_{c_\ell(\mathbf{x})}$ from Lemma B.3 to Definition B.5 gives

$$\mathbf{u}'_{c_i,c_j} = \frac{1}{N_{c_i,c_j}} \sum_{\mathbf{x} \in \mathcal{D}_{c_i,c_j}} \left[ f(\mathbf{x}) - \mathbf{f} \right] \tag{12}$$

$$= \frac{1}{N_{c_i,c_j}} \sum_{\mathbf{x} \in \mathcal{D}_{c_i,c_j}} \left[ \mathbf{f} + \sum_{\ell=1}^c \mathbf{u}'_{c_\ell(\mathbf{x})} - \mathbf{f} \right] \tag{13}$$

$$= \frac{1}{N_{c_i,c_j}} \sum_{\mathbf{x} \in \mathcal{D}_{c_i,c_j}} \sum_{\ell=1}^c \mathbf{u}'_{c_\ell(\mathbf{x})}. \tag{14}$$

For every $\mathbf{x} \in \mathcal{D}_{c_i,c_j}$ we have $c_i(\mathbf{x}) = c_i$ and $c_j(\mathbf{x}) = c_j$. Hence the terms with $\ell = i$ and $\ell = j$ contribute exactly $\mathbf{u}'_{c_i}$ and $\mathbf{u}'_{c_j}$, respectively.

For any $\ell \notin \{i, j\}$ each value $c'_\ell \in \mathcal{C}_\ell$ occurs equally often inside $\mathcal{D}_{c_i,c_j}$. Therefore

$$\frac{1}{N_{c_i,c_j}} \sum_{\mathbf{x} \in \mathcal{D}_{c_i,c_j}} \mathbf{u}'_{c_\ell(\mathbf{x})} = \frac{1}{|\mathcal{C}_\ell|} \sum_{c'_\ell \in \mathcal{C}_\ell} \mathbf{u}'_{c'_\ell} = \mathbf{0}, \quad \text{by Lemma B.4.}$$

Collecting all contributions we obtain the desired identity

$$\mathbf{u}'_{c_i,c_j} = \mathbf{u}'_{c_i} + \mathbf{u}'_{c_j}.$$

$\square$

We now establish our main theoretical result on the minimal data requirements for compositional generalization. The derivations from the Lemmas above are appropriate under the assumption of a balanced training set. Due to the unlikely nature of certain concept combinations (as described in the $(n, k)$ framework), the main challenge is identifying the factors under such a setting.

**Proposition B.7** (Minimal compositional learning). *Let $f : \mathcal{X} \to \mathbb{R}^d$ be a feature extractor with linearly factored concept embeddings over $\mathcal{C}$. Let $\{\mathbf{u}_{c_1^1}, \ldots, \mathbf{u}_{c_1^n}\}$ and $\{\mathbf{u}_{c_2^1}, \ldots, \mathbf{u}_{c_2^n}\}$ be the concept vectors for the first and second concepts respectively, where their joint span has dimension $2n - 1$. Suppose we only observe joint representations for concept combinations $c_i, c_j \in \{1, \ldots, n\}$. Then $k = 2$ combinations per concept value suffice to learn a linear classifier that perfectly generalizes to all $(n - k) \cdot n$ unseen combinations.*

*Proof.* The proof proceeds in three steps: (1) showing that joint factored embeddings are identifiable from training data, (2) showing that the system of linear equations has full rank with $2n$ equations and $2n$ unknowns, and (3) showing that optimal classifiers can be constructed via orthogonal projections.

**Part 1: Identifying joint factored embeddings $\mathbf{u}_{c_1^i, c_2^j}$.**

We assume $k = 2$ for simplicity, but the same applies for higher $k$. First, note that we observe the following combinations:

$$\mathcal{C}_{\text{train}} = \{(i, i) : i \in [n]\} \cup \{(i, i+1) : i \in [n-1]\} \cup \{(n, 1)\} \tag{15}$$
$$= \{(1, 1), (2, 2), \ldots, (n, n)\} \cup \{(1, 2), (2, 3), \ldots, (n-1, n)\} \cup \{(n, 1)\} \tag{16}$$

with $|\mathcal{C}_{\text{train}}| = 2n$ total combinations. This dataset is restricted to the combinations in $\mathcal{C}_{\text{train}}$, but varies in other concepts. We denote this dataset as $\mathcal{D}_{\text{train}} := \{(c_1, c_2, \mathbf{x}) : (c_1, c_2) \in \mathcal{C}_{\text{train}}, \mathbf{x} \in \mathcal{X}\}$.

We aim to show that the average embedding over the training set, $\bar{\mathbf{u}}_{\text{train}}$, equals the global mean embedding $\mathbf{f}$ (as defined in the proof of Lemma B.4). Let $\mathcal{D}_{i,j} \subset \mathcal{D}_{\text{train}}$ be the subset of training samples for the specific concept combination $(i, j)$. To see the importance of this, note that

$$\mathbf{u}'_{c_1^i, c_2^j} = \mathbf{u}'_{c_1^i} + \mathbf{u}'_{c_2^j}. \tag{17}$$

By Definition B.5, given some observations of concept values $c_1^i$ and $c_2^j$, the pairwise joint embedding $\mathbf{u}'_{c_1^i, c_2^j}$ is the average of the embeddings of the training samples for the combination $(i, j)$ shifted by the global mean embedding $\mathbf{f}$. Consider the mean embedding over the training set

$$\bar{\mathbf{u}}_{\text{train}} := \frac{1}{|\mathcal{D}_{\text{train}}|} \sum_{\mathbf{x} \in \mathcal{D}_{\text{train}}} f(\mathbf{x}). \tag{18}$$

We now show that $\mathbf{f} = \bar{\mathbf{u}}_{\text{train}}$.

Under the assumption of a balanced training set where each combination $(i, j) \in \mathcal{C}_{\text{train}}$ has the same number of samples, we can define the mean embedding for each combination as:

$$\bar{\mathbf{f}}_{i,j} := \frac{1}{|\mathcal{D}_{i,j}|} \sum_{\mathbf{x} \in \mathcal{D}_{i,j}} f(\mathbf{x}).$$

The overall training mean is then:

$$\bar{\mathbf{u}}_{\text{train}} := \frac{1}{|\mathcal{D}_{\text{train}}|} \sum_{\mathbf{x} \in \mathcal{D}_{\text{train}}} f(\mathbf{x}) \tag{19}$$

$$= \frac{1}{2n} \left( \sum_{i=1}^{n} \bar{\mathbf{f}}_{i,i} + \sum_{i=1}^{n-1} \bar{\mathbf{f}}_{i,i+1} + \bar{\mathbf{f}}_{n,1} \right) \tag{20}$$

$$= \frac{1}{2n} \left( \sum_{i=1}^{n} (\mathbf{f} + \mathbf{u}'_{c_1^i} + \mathbf{u}'_{c_2^i}) + \sum_{i=1}^{n-1} (\mathbf{f} + \mathbf{u}'_{c_1^i} + \mathbf{u}'_{c_2^{i+1}}) + (\mathbf{f} + \mathbf{u}'_{c_1^n} + \mathbf{u}'_{c_2^1}) \right) \tag{21}$$

$$= \frac{1}{2n} \left( 2n\mathbf{f} + 2\sum_{i=1}^{n} \mathbf{u}'_{c_1^i} + 2\sum_{i=1}^{n} \mathbf{u}'_{c_2^i} \right) \tag{22}$$

$$= \frac{1}{2n} (2n\mathbf{f} + 2 \cdot \mathbf{0} + 2 \cdot \mathbf{0}) \quad \text{(by Lemma B.4)} \tag{23}$$

$$= \mathbf{f} \tag{24}$$

Thus, we can identify the factored representations $\mathbf{u}_{c_1^i, c_2^j}$ for each concept value combination $i, j \in [n]$ from the training data since the average representation over the training data under our training dataset is the global mean embedding $\mathbf{f}$. With this, we can compute $\mathbf{u}'_{c_1^i, c_2^j}$ for $2n$ combinations.

**Part 2: Identifying the individual factored representations $\mathbf{u}_{c_1^i}$ and $\mathbf{u}_{c_2^i}$ for each concept value $i \in [n]$.**

Consider a training set with exactly two combinations per concept value. By the linear factorization property, for any combination $(i, j)$ in our training set, we have: $\mathbf{u}'_{c_1^i, c_2^j} = \mathbf{u}'_{c_1^i} + \mathbf{u}'_{c_2^j}$, where $c_1^i$ denotes value $i$ for the first concept and $c_2^j$ denotes value $j$ for the second concept.

Let $\mathbf{U}_1, \mathbf{U}_2 \in \mathbb{R}^{d \times n}$ be matrices whose columns are the unknown factored representations $\mathbf{u}'_{c_1^i}$ and $\mathbf{u}'_{c_2^i}$ respectively for $i \in [n]$. Let $\mathbf{V} \in \mathbb{R}^{d \times 2n}$ be the matrix of observed pairwise joint embeddings $\mathbf{u}'_{c_1^i, c_2^j}$ for the $2n$ training combinations. The system of equations can be written as:

$$\underbrace{\begin{bmatrix} \mathbf{u}'_{c_1^1, c_2^1} \\ \mathbf{u}'_{c_1^2, c_2^2} \\ \vdots \\ \mathbf{u}'_{c_1^n, c_2^n} \\ \hline \mathbf{u}'_{c_1^1, c_2^2} \\ \mathbf{u}'_{c_1^2, c_2^3} \\ \vdots \\ \mathbf{u}'_{c_1^{n-1}, c_2^n} \\ \mathbf{u}'_{c_1^n, c_2^1} \end{bmatrix}}_{\mathbf{V}} = \left[ \begin{array}{cccc|cccc} 1 & 0 & \cdots & 0 & 1 & 0 & \cdots & 0 \\ 0 & 1 & \cdots & 0 & 0 & 1 & \cdots & 0 \\ \vdots & \vdots & \ddots & \vdots & \vdots & \vdots & \ddots & \vdots \\ 0 & 0 & \cdots & 1 & 0 & 0 & \cdots & 1 \\ \hline 1 & 0 & \cdots & 0 & 0 & 1 & \cdots & 0 & 0 \\ 0 & 1 & \cdots & 0 & 0 & 0 & \ddots & 0 & 0 \\ \vdots & \vdots & \ddots & \vdots & \vdots & \vdots & \ddots & \vdots & \vdots \\ 0 & 0 & \cdots & 1 & 0 & 0 & \cdots & 1 & 0 \\ 1 & 0 & \cdots & 0 & 0 & 0 & \cdots & 0 & 1 \end{array} \right] \begin{bmatrix} \mathbf{u}'_{c_1^1} \\ \mathbf{u}'_{c_1^2} \\ \vdots \\ \mathbf{u}'_{c_1^n} \\ \mathbf{u}'_{c_2^1} \\ \mathbf{u}'_{c_2^2} \\ \vdots \\ \mathbf{u}'_{c_2^n} \end{bmatrix} \begin{bmatrix} \mathbf{U}_1 \\ \hline \mathbf{U}_2 \end{bmatrix} \tag{25}$$

We note that this system is full rank, as the design matrix has linearly independent rows. The first block of rows corresponds to the diagonal combinations $(i, i)$, while the second block corresponds to cyclic combinations $(i, i + 1)$ (with wraparound from $n$ to $1$). These form distinct patterns that ensure linear independence.

Given this full rank system with $2n$ equations and $2n$ unknowns (the factored representations $\mathbf{u}_{c_1^i}$ and $\mathbf{u}'_{c_2^i}$ for each concept value), we can uniquely solve for the factored representations. For $k > 2$ combinations per concept value, we get more equations while maintaining the same number of unknowns, making the system overdetermined and the solution more robust.

Once we recover these factored representations, we can compute $\mathbf{u}'_{c_1^i, c_2^j} = \mathbf{u}'_{c_1^i} + \mathbf{u}'_{c_2^j}$ for any combination $(i, j)$, including the $(n-2)n$ unseen ones.

**Part 3: Optimality of classifiers.** To show that we can construct classifiers that provable generalize to novel combinations, we simply note that by assumption no concept representation is within the span of remaining representations. As such, given $U_1 := \text{span}(\{\mathbf{u}'_{c_1^i}\}_{i=1}^{|\mathcal{C}_1|})$, and $U_2 := \text{span}(\{\mathbf{u}'_{c_2^i}\}_{i=1}^{|\mathcal{C}_2|})$, such that $\dim(U_1) = |\mathcal{C}_1| - 1$ and $\dim(U_2) = |\mathcal{C}_2| - 1$ and $U_1 \cap U_2 = \{0\}$, any vector $\mathbf{w}$ in their joint span can be uniquely decomposed as $\mathbf{w} = \mathbf{u}_1 + \mathbf{u}_2$ where $\mathbf{u}_1 \in U_1$, $\mathbf{u}_2 \in U_2$ and $\mathbf{u}_1 \perp \mathbf{u}_2$. This allows us to construct projection matrices $P_{U_1}$ and $P_{U_2}$ onto these orthogonal subspaces, which can then be used to build optimal classifiers by projecting input features onto the respective concept subspaces.

$\square$

### B.1. Algorithmic recovery of factored representations

We provide a constructive algorithm for recovering factored concept representations from limited available training combinations in Algorithm 1.

---

**Algorithm 1** Recovering factored concept representations for $k = 2$ concepts

---

**Require:** Training dataset $\mathcal{D}_{\text{train}}$ where each individual concept appears in at least 2 different combinations ($k \geq 2$)
**Require:** Feature extractor $f : \mathcal{X} \to \mathbb{R}^d$
**Ensure:** Factored concept representations $\{\mathbf{u}'_{c_1^i}\}_{i=1}^n$, $\{\mathbf{u}'_{c_2^i}\}_{i=1}^n$

1: Compute global mean embedding: $\mathbf{f}_d \leftarrow \frac{1}{|\mathcal{D}_{\text{train}}|} \sum_{\mathbf{x} \in \mathcal{D}_{\text{train}}} f(\mathbf{x})_d$ for each dimension $d$
2: **for** $d = 1$ to $d$ **do**
3:      Initialize design matrix $\mathbf{A} \in \mathbb{R}^{2n \times 2n}$ based on observed combinations
4:      Initialize $\mathbf{v} \in \mathbb{R}^{2n}$ to store joint embeddings for dimension $d$
5:      $row \leftarrow 1$
6:      **for** each combination $(i, j)$ in training set **do**
7:          $\mathbf{u}'_{c_1^i, c_2^j} \leftarrow \frac{1}{|\{\mathbf{x}:c(\mathbf{x})_1=i, c(\mathbf{x})_2=j\}|} \sum_{\mathbf{x}:c(\mathbf{x})_1=i, c(\mathbf{x})_2=j} f(\mathbf{x})_d - \mathbf{f}_d$
8:          Store $\mathbf{u}'_{c_1^i, c_2^j}$ in position $row$ of $\mathbf{v}$
9:          Update row $row$ of $\mathbf{A}$ with indicators for concepts $i$ and $j$
10:          $row \leftarrow row + 1$
11:      **end for**
12:      Solve system $\mathbf{A} \begin{bmatrix} \mathbf{u}'_1 \\ \mathbf{u}'_2 \end{bmatrix} = \mathbf{v}$ for dimension $d$
13:      Store solutions in $\{u'_{c_1^i}\}_{i=1}^n$, $\{u'_{c_2^i}\}_{i=1}^n$ at dimension $d$
14: **end for**
15: **return** $\{\mathbf{u}'_{c_1^i}\}_{i=1}^n$, $\{\mathbf{u}'_{c_2^i}\}_{i=1}^n$

---

# C. Additional experimental results

This section presents supplementary experimental findings.

### C.1. From-scratch model performance

Figure 9 summarizes how out-of-distribution accuracy varies with the number of concept classes and the number of training combinations per class across four datasets. In all cases, increasing concept diversity (number of classes) is associated with higher compositional generalization performance, even when the number of training combinations per class is held fixed.

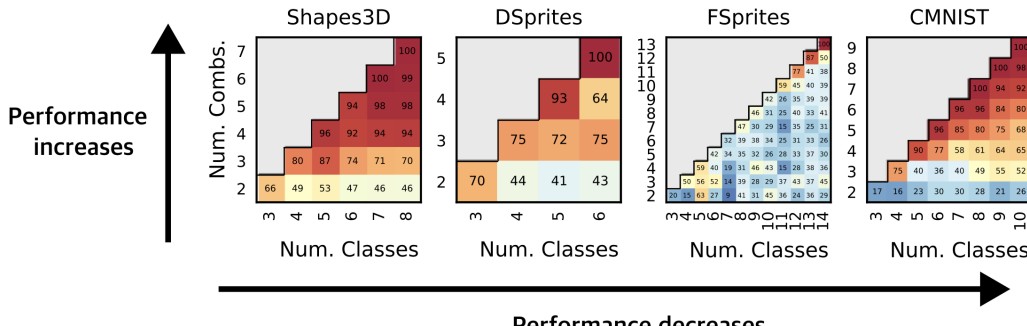

Figure 9: **Performance scaling with concept diversity.** OOD accuracies across four datasets: Shapes3D, dSprites, FSprites, and Colored-MNIST. Each heatmap shows performance for different combinations of concept values ($n$) and seen combinations ($k$) per concept value. Increasing concept diversity (higher $n$) consistently improves generalization performance across all datasets, even when the number of training combinations per concept remains fixed.

## C.2. Pre-trained model probing results

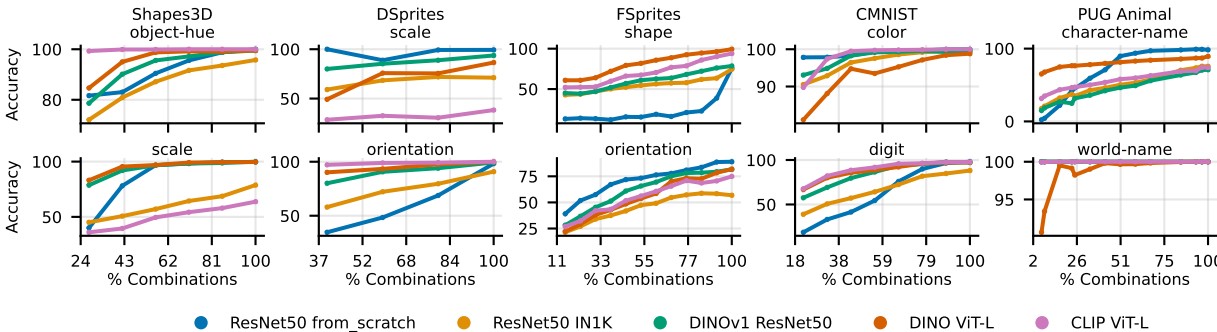

Figure 10: **Compositional generalization in pre-trained models.** Heatmaps show out-of-distribution accuracy for different combinations of $n$ (concept values) and $k$ (training combinations) across datasets. Darker colors indicate higher accuracy. Pre-trained models exhibit improved generalization with increased concept diversity, mirroring the pattern observed in from-scratch training.

To systematically probe compositional generalization in pre-trained vision models, we evaluated a range of architectures, including ResNet50 (from scratch and ImageNet pre-trained), DINOv1, DINO ViT-L, and CLIP ViT-L across several datasets and concept axes, as shown in Figure 10.

## C.3. MPI3D dataset results

To validate our findings on real-world datasets, we conduct experiments on the MPI3D dataset (Gondal et al., 2019), which contains photographs of 3D scenes with systematic concept variations.

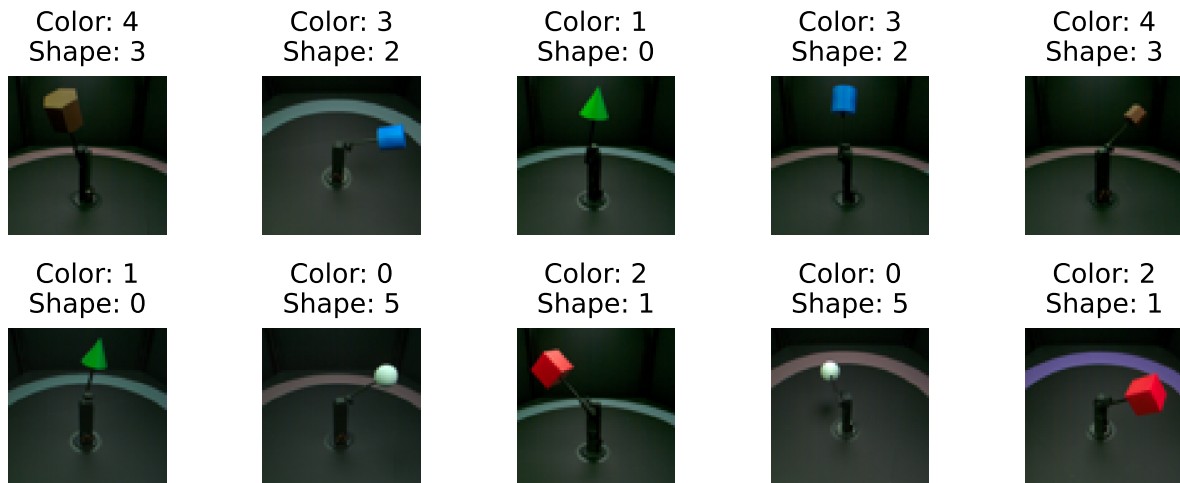

Figure 11: Sample images from the MPI3D dataset (Gondal et al., 2019). The dataset contains real-world images of objects with varying properties like color, shape, size and camera viewpoint. Examples from the testing set of $n = 6, k = 5$ are shown.

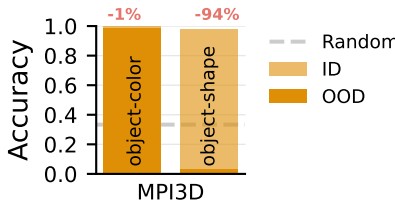

Figure 12: Accuracy comparison for $n = 3, k = 2$ using ResNet-50. As shown in the main text, compositional generalization is difficult: the model struggles to generalize to the object-shape concept.

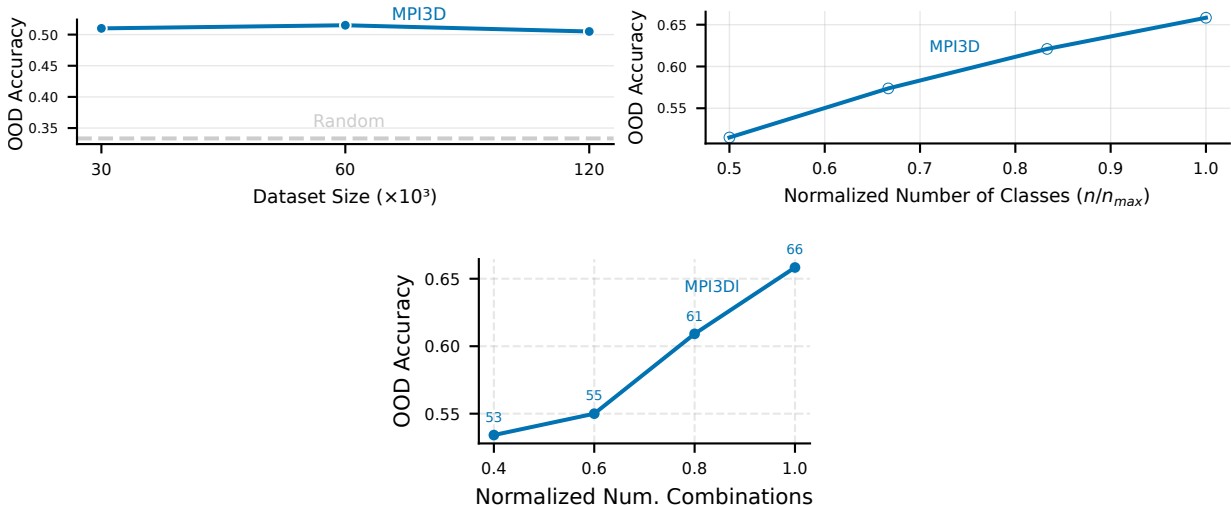

Figure 13: Compositional generalization only improves with data diversity, not data quantity. Top left: Under few training combinations (n = 3, k = 2), compositional generalization does not benefit from more ID data. The remaining plots show compositional generalization improving with more diverse training combinations: when the number of classes increases (top right), and when the number of training combinations increases (bottom left)

These results provide strong evidence that compositional generalization benefits specifically from *diversity* in concept

combinations rather than mere quantity of training data.

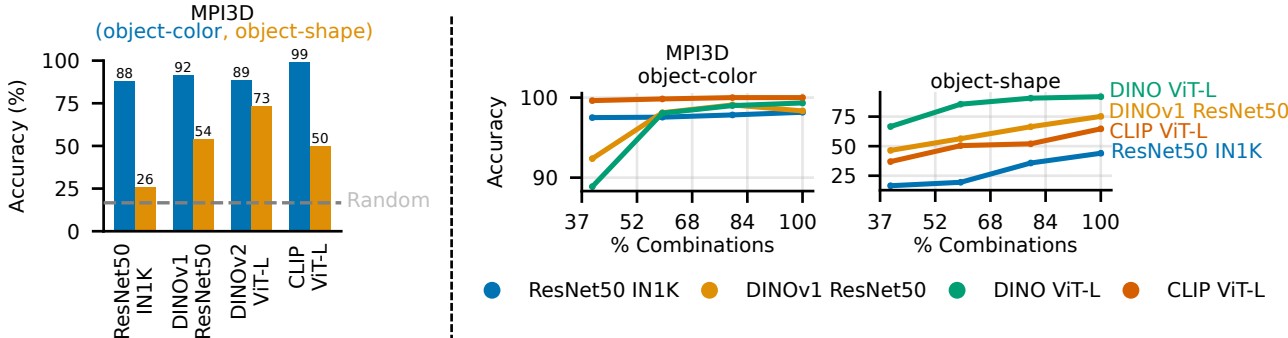

Figure 14: Evaluating pre-trained vision models on MPI3D. Left: Accuracy comparison for classifiers constructed under linear factorization. All models show near-perfect accuracy on the color concept, while shape concept performance is worse. Right: Probing results using linear and non-linear probes.

## C.4. Comparison of different probe configurations

We present a detailed comparison of probe results across different model architectures and probe types. Table 1 reports the accuracy of both linear and non-linear (two-layer MLP) probes on the FSprites dataset for several pre-trained models. Notably, non-linear probes generally yield higher accuracy than linear probes, especially for models like DINO ResNet-50 and DINO ViT-Large, indicating that some compositional information is not linearly accessible in the representations. However, for ResNet-50 and CLIP ViT-Large, the difference between linear and non-linear probe performance is smaller, suggesting that their representations are more linearly separable for the evaluated concepts.

Table 1: **Linear and non-linear probing results.** Comparison between linear probes and two-layer MLPs `[512,512]` as the observed percentage of combinations on FSprites dataset. Results show the accuracy in the form of linear / non-linear probing for different pre-trained models.

| Model | 25% | 50% | 75% | 93% |
|---|---|---|---|---|
| ResNet-50 ImageNet | 0.59 / 0.55 | 0.67 / 0.65 | 0.75 / 0.75 | 0.79 / 0.82 |
| DINO ResNet-50 | 0.60 / 0.67 | 0.71 / 0.80 | 0.76 / 0.88 | 0.80 / 0.92 |
| DINO ViT-Large | 0.68 / 0.70 | 0.78 / 0.83 | 0.84 / 0.91 | 0.86 / 0.95 |
| CLIP ViT-Large | 0.61 / 0.64 | 0.70 / 0.74 | 0.75 / 0.79 | 0.76 / 0.84 |

## C.5. Architecture comparisons

We provide detailed comparisons between different neural architectures to validate our choice of ResNet-50 as the primary baseline.

A comprehensive hyper-parameter sweep was conducted for the vision transformer (ViT), varying patch size ($\in \{8, 16\}$), depth ($\in \{4, 6, 8\}$), width ($\in \{384, 512\}$), number of heads ($\in \{8, 12\}$), MLP width ($\in \{384, 512\}$), and learning rate ($\in \{3{\times}10^{-4}, 1{\times}10^{-4}, 3{\times}10^{-5}\}$). Across all configurations, ViT does not outperform a scratch-trained ResNet-50 in OOD generalisation. Both models achieve comparable in-distribution accuracy (99.7%), but the ResNet-50 baseline consistently yields higher OOD performance across datasets and diversity regimes. Table 2 summarises these results.

Table 2: Accuracy of ResNet50 and ViT models trained from scratch.

| Dataset | Model | % Combinations | $k/n$ | Training samples ($\times 10^3$) | OOD Acc. |
|---------|-------|----------------|-------|-------------------------------|----------|
| CMNIST | ResNet-50 | 80 | 8 / 10 | 60 | 95.1 |
| | ViT | 80 | 8 / 10 | 60 | 94.5 |
| | ResNet-50 | 40 | 4 / 10 | 60 | 66.0 |
| | ViT | 40 | 4 / 10 | 60 | 71.0 |
| FunnySprites | ResNet-50 | 92 | 13 / 14 | 60 | 80.1 |
| | ViT | 92 | 13 / 14 | 60 | 66.0 |
| | ViT$^\dagger$ | 92 | 13 / 14 | 120 | 57.3 |

# D. Dataset details and examples

This section provides comprehensive information about all datasets used in our experiments, including detailed descriptions and visual examples.

Table 3: **Overview of experimental datasets.** Each dataset provides controlled variations along two primary concept dimensions, enabling systematic study of compositional generalization.

| Dataset | Primary Concepts ($\mathcal{C}_1, \mathcal{C}_2$) | Concept Values |
|---------|--------------------------------------------------|----------------|
| PUG | Animal type, Background type | 60 each |
| Shapes3D | Scale, Object hue | 8 each |
| dSprites | Scale, Orientation | 6 each |
| FunnySprites | Shape, Color | 14 each |
| Colored-MNIST | Digit, Color | 10 each |
| MPI3D | Object shape, Object color | Variable |

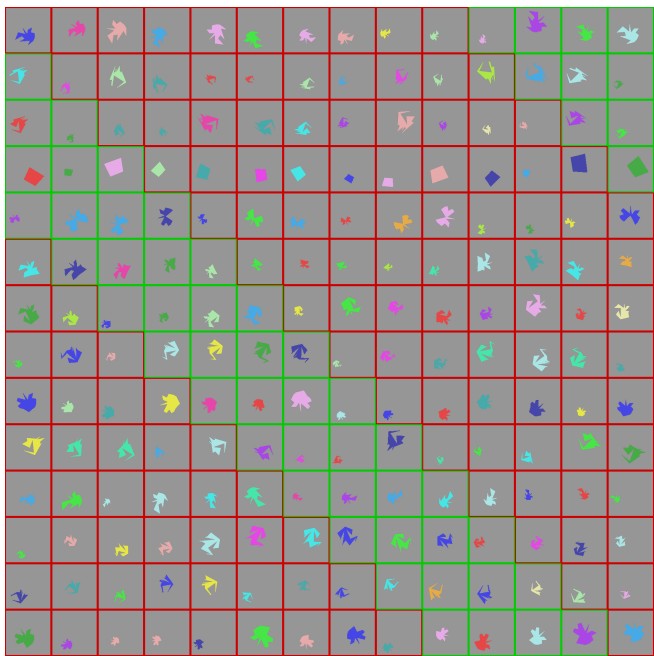

Figure 15: **FunnySprites dataset examples.** Shape and orientation variations for $n = 14$ concept values with $k = 2$ training combinations. Each sprite is generated by connecting traced points to form unique geometric shapes, providing a challenging test for compositional generalization.

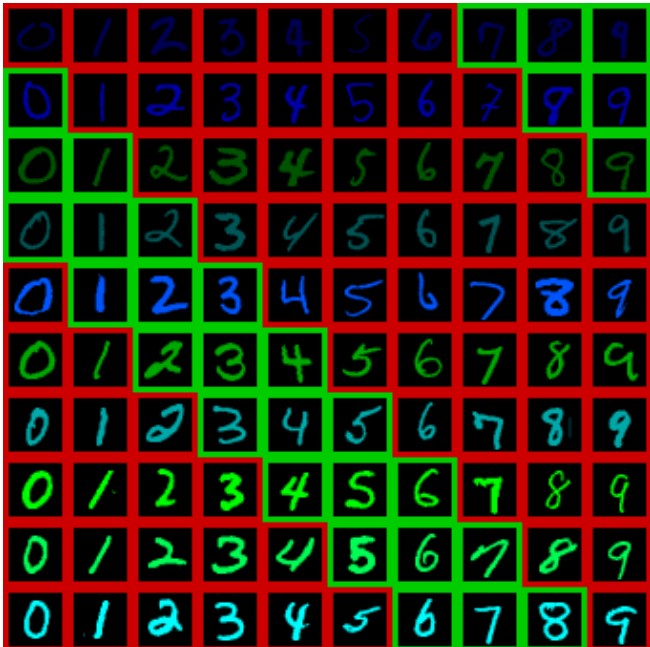

Figure 16: **Colored-MNIST examples.** Digit and color combinations for $n = 10$ values with $k = 3$ training combinations. This dataset combines the MNIST digits with color variations to test compositional understanding of shape and color attributes.

We introduce the Funny Sprites dataset, an OOD dataset designed to test models' ability to generalize to previously unseen shape combinations. The dataset consists of sprites traced from 5-15 points on a 128x128 pixel grid, creating a diverse set of abstract geometric shapes.

