# OpenReview forum: "Does Data Scaling Lead to Visual Compositional Generalization?"
_ICML.cc/2025/Conference — ICML 2025 poster_

### Official Review · Reviewer_TQRD · 2025-03-10

**Overall Recommendation:** 4

**Summary:**

The article investigates how the compositional generalization capacity of vision models is related to the scale and quality of the training dataset. The authors main contribution is the proposal of a systematic way to manipulate training and testing dataset splits in order to evaluate how data diversity promotes generalization. Additionally, they introduce a novel dataset composed on abstract shapes called Funny-Sprites which provides a more comprehensive benchmark than the more standard dSprites. The authors perform a thorough evaluation using both pre-trained models and models trained from scratch. Their findings seem to indicate that data diversity is crucial for compositional generalization, yet cannot achieve perfect compositionally at the scales they investigated.

**Claims And Evidence:**

In general, the authors make a really good effort at substantiating their claims with solid empirical evidence. They evaluate several variations of their test protocol, varying the number of values each concept can take, and the number of combinations each value can appear in. This allows them to manipulate both the diversity of the data and, potentially, the qualitative nature of the missing combinations (though they don't really explore this angle too much).

They do not test a large variety of models when training from scratch (only ResNet-50 is reported), but they claim to have used other baselines in the text. It would  be good to include some of these results, even if they are not used in all the conditions. One way to justify not including all test conditions could be to start with simple ones and, should they fare worse than ResNet-50, exclude them from subsequent evaluations.

**Essential References Not Discussed:**

No that I can remember. The authors are very thorough in their discussion of related work.

**Experimental Designs Or Analyses:**

Yes, I did. They appear correct to me.

**Methods And Evaluation Criteria:**

For the most part, yes. The datasets used are standard in the compositional generalization literature. Additionally, they introduced a new dataset called Funny-Sprites which contains abstract, non-regular shapes. They also analyze the learned representations using several metrics to understand how they are structures (i.e. are they more linear, can concept values be decoded, do they lead to better generalization).

**Other Comments Or Suggestions:**

My main suggestion is regarding the use of the term compositionally. Throughout the paper the authors use it as equivalent to orthogonal/linear representations, yet this is not exactly correct. Rather, having a representation as the one in equation 25 is one way, but not the only way, to implement a compositional representation. Indeed, it is a very popular view of compositionally in DL. However, it is perfectly valid to imagine a representation that does not conform to such notions of linearity/orthogonality, but is still compositional. The only requirement is that whatever mechanism is doing the composition can handle such representation. Thus, I would suggest that the authors make it clear that this is the implementation of compositionally they are exploring. Moreover, the main issue is whether such a representation is stable in the models being studied.

In fact, the author's own findings contradict that claim that these are compositional representations, since they fail to generalize systematically to unseen combinations, a fact that authors point to briefly when they state that they are not "ideal". Note that in the Philosophy of Mind literature compositionality implies systematicity; thus if a model is not systematic it cannot be compositional (or "ideally compositional", see Fodor and Pylyshyn). Thus the representations must either be unstable (they deviate from the expected position in latent space for that combination) or such a representation is not enough to ensure compositionality.

**Other Strengths And Weaknesses:**

The authors explore an important question: how does data diversity influence compositionality, which is important since one popular claim is that scaling data is enough to achieve strong generalization in AI systems. These findings suggest that this is in fact not the case.

**Questions For Authors:**

What happens if in the analysis in section 4.2, instead of training the decoder and determining PCA directions with both training and testing combinations the authors only se the training ones and predict/project the test combinations with the resulting model/PCA? Do the test combinations clearly separate from the training ones? Is the nice lattice structure clearly observed break? I would like to see this analysis.

**Relation To Broader Scientific Literature:**

The paper is well situated within the literature. The authors make a substantial effort to cover the related literature and identify where their contribution lies, and in my opinion they succeed at this.

**Theoretical Claims:**

I only skimmed through them since they are in the appendix, but looked fine. The claims at least make intuitive sense.

---

> ### Author Rebuttal · Authors · 2025-04-01
>
> We appreciate your thorough review. We will incorporate the feedback into the updated manuscript.
>
> **They do not test a large variety of models when training from scratch (only ResNet-50 is reported), but they claim to have used other baselines in the text. It would be good to include some of these results, even if they are not used in all the conditions. One way to justify not including all test conditions could be to start with simple ones and, should they fare worse than ResNet-50, exclude them from subsequent evaluations.**
>
> Thank you for this suggestion. We agree that including results from additional models would strengthen the completeness of our empirical analysis. We include some of the results of vision transformers and compare them with ResNet50 here by varying the % of combinations for CMNIST (since the model achieves comparable results with 80% of combinations); additionally, we highlight that even under increased number of training points, vision transformer still underperforms (all experiments assumed 60,000 datapoints, except for the last row):
>
> | Dataset       | Model     | % Combinations ($k/n$) | OOD Accuracy |                                   |
> |--------------|-----------|------------------|--------------|-------------------------------------------|
> | CMNIST       | ResNet50  | 80% (8/10)        | 95.1         |                                           |
> |              | ViT       | 80% (8/10)        | 94.5         |                  |
> |              | ResNet50  | 40% (4/10)        | 66.0         |      |
> |              | ViT       | 40% (4/10)        | 71.0         |      |
> | FunnySprites | ResNet50  | 92% (13/14)       | 80.1         |
> |              | ViT       | 92% (13/14)       | 66.0         |                                           |
> |              | ViT (120K data)       | 92% (13/14)       | 57.3         |  |
>
> We attempted to be as charitable to ViT architectures as possible: we varied the patch size (8 and 16), ViT depth (4, 6, 8), width of each layer (384, 512), number of heads (8, 12), MLP width (384, 512), and learning rate ($1e{-}5$, $1e{-}4$, $1e{-}3$). In the table above, we report the best-performing configuration for each setting. All the models were able to achieve 99.7% accuracy on the ID data, but ViT typically underperformed on the testing data.
>
> **My main suggestion is regarding the use of the term compositionality.**
>
> Thank you for raising this point. We agree that the term “compositionality” may not have been the most precise description for the notion we are exploring. Our motivation for using it stemmed from the natural emergence of linear factorization in the from-scratch setting with increased data diversity. However, we acknowledge that compositional structure does not have to be linear. Instead, we propose to use the term “linear factorization” as a specific instantiation of compositionality. We will clarify this in the text.
>
> **In fact, the authors’ own findings contradict the claim that these are compositional representations, since they fail to generalize systematically to unseen combinations. [...] Thus the representations must either be unstable or such a representation is not enough to ensure compositionality.**
>
> Our goal is not to claim that the representations are fully compositional, but rather to study to which extent they are compositional, instantiated in our case as exhibiting a linear factorization. The failure to generalize to unseen combinations highlights the limitations of this structure in the models we study. If our writing gave the impression that we claim full compositionality, we’d be happy to clarify--please do let us know if a particular phrasing seemed misleading.
>
> **What happens if in the analysis in Section 4.2, instead of training the decoder and determining PCA directions with both training and testing combinations, the authors only use the training ones and project the test combinations with the resulting model/PCA? Do the test combinations clearly separate from the training ones? Is the nice lattice structure observed in the projections disrupted?**
>
> The displayed points in Section 4.2 are from the OOD data, i.e., unseen combinations. Specifically, we sample a 2×2 grid of combinations that share concept values (i.e., (i, j), (i, k), (l, j), (l, k)), all from the test set. We will clarify this procedure in text.
>
> Regarding the suggestion to perform PCA only on the training data and project the test points using the resulting model: in this case, the lattice structure is not preserved. Intuitively, this is because combinations of concept values can span up to a $2n - 1$-dimensional space. Even if the training data admits a linear factorization, projecting unseen combinations using a PCA fit on the training data alone does not guarantee that the structure will be preserved since the model is trained to classify concepts. This is the reason why we used a smaller set of combinations for visualization purposes.

---

### Official Review · Reviewer_Nv93 · 2025-03-13

**Overall Recommendation:** 3

**Summary:**

This paper studies whether compositional generalization emerges in vision models trained from scratch and large-scale datasets. The authors design simple experiments where two factors, i.e., shape and color, control the dataset. The authors explore different levels of compositions in the training set and offer some discussions based on the results.

**Claims And Evidence:**

- “large pretrained models” can be very misleading. I would suggest the authors always emphasize “vision models pretrained with large-scale datasets”.
- L90 “settings where no restrictions on the scale of compositions are imposed” is not very clear what it means exactly.
- It is challenging to see the relation between this work and others mentioned in related work, especially in the last four paragraphs. This further makes it difficult to judge the contribution of this work.

**Essential References Not Discussed:**

- Zerroug et al., A Benchmark for Compositional Visual Reasoning, NeurIPS 2022, shared a highly similar topic.
- Zhou et al., Data Factors for Better Compositional Generalization, EMNLP 2023. Although the paper focuses on NLP domain, it also discusses compositionality from a data perspective.
- Stone et al., Teaching Compositionality to CNNs, CVPR 2017 has demonstrated that training from scratch offers better compositionality generalization.

**Experimental Designs Or Analyses:**

- L186, “contains additional concept dimensions (like position, orientation, or background) with |Ci| possible values each. For instance, in a dataset with two additional dimensions having |C1| = 8 and |C2| = 12 possible values respectively”, it is unclear why authors introduce more variants to the compositions while claiming that the paper limits its compositionality as 2 for simplicity.

**Methods And Evaluation Criteria:**

- “ID data” is not explained at all in the paper and is used many times.

**Other Comments Or Suggestions:**

L95, “Vision-language models face specific compositional challenge”, would be great to explain a bit more in texts.

L129, “face their own limitations in handling complex visual compositions.”, it would be great to explain a bit more in texts.

L215, "We study this through two main sets of experiments: pre-training, where we train models from scratch (Section 4), and evaluating pre-trained foundation models’ (FM) compositional abilities…” is very confusing and is probably just a typo? I suggest just rephrasing “pre-training, where we train models from scratch” as “trained from scratch”.

**Other Strengths And Weaknesses:**

**Strengths**

- The introduction is well-written and easy to follow.
- The studied topic is very important to human-like AI development.

**Weakness:**
- “in this work, we focus specifically on pairwise compositions” Is there any reference to follow and justify whether the simplification is reasonable and effective for its extension?

My major concerns are:
 1. It is unclear what the difference is in the empirical findings from existing work.
 2. It is unclear whether the simple experimental setting in this paper can scale up to compositions with various compositionality dimensions in the real world.

**Questions For Authors:**

Please see comments and suggestions.

**Relation To Broader Scientific Literature:**

The key contributions of this paper could facilitate compositionality learning, a core component of human-like AI.

**Theoretical Claims:**

yes

---

> ### Author Rebuttal · Authors · 2025-04-01
>
> Thank you for your thorough review. We will incorporate these suggestions in the final version of the manuscript.
>
> **"large pretrained models" can be very misleading**
>
> We agree about emphasizing the vision aspect. However, our models are large within the vision domain—for example, DINO ViT-L contains 300M parameters. Would it be clearer if we added a footnote explaining our criterion for "large-scale models" in the vision context?
>
> **L90 is not very clear what it means**
>
> What we meant by this is that prior work usually considers fixed compositionality settings where the scale of compositions is fixed.
>
> **It is challenging to see the relation between this work and others mentioned in related work, especially in the last four paragraphs.**
>
> We aimed to cover prior work relevant to (compositional) generalization and vision, and highlight challenges. In brief:
> - *Compositionality and vision models:* We discuss existing limitations in vision models, e.g. under spurious correlations and how it affects generalization.
> - *Scaling and emergent abilities:* Prior work shows scaling can improve performance in (compositional) generalization tasks. We build on this by systematically varying data and model scale across multiple axes.
> - *Improving compositionality:* While others introduce architectural or loss-based modifications, we evaluate whether modern scaling practices alone suffice for compositional generalization.
> - *Representation learning:* Our approach focuses on the structure of learned representations--specifically, how they support generalization to unseen compositions.
>
> We will revise the text to more clearly explain how our work relates to prior work.
>
> **"ID data" is not explained.**
>
> We agree that this terminology is confusing without an explicit mention of it. By ID data we are referring to the data sampled from the training region of combinations defined by the $(n, k)$ framework we introduced.
>
> **L186: it is unclear why authors introduce more variants to the compositions while claiming that the paper limits its compositionality as 2 for simplicity.** & **Is there any reference to follow and justify whether the simplification is reasonable and effective for its extension?**
>
> We wanted to emphasize that the total dataset size is not restricted by the two concepts we study. While the specific pair of concepts we focus on is limited in size, other concepts in the data can vary freely. This allows us to study a more realistic setting for compositionality. For example, in the real world, when we restrict the pair (color, animal), we know that blue and green pandas aren't common in datasets. However, the pairs that are present (e.g., white pandas) can still vary along many other dimensions, such as size, pose, location, and other attributes.
>
> Additionally, we focus on the two-concept case because we believe it is the simplest compositional setup that still presents challenges for vision models. Given that the total number of concepts is greater than 2, applying further restrictions would only make the task harder. Therefore, we view the two-concept focus as a reasonable simplification.
>
> **Zerroug et al. (2022), Zhou et al. (2023), and Stone et al. (2017) are essential related works not discussed.** & **It is unclear what the difference is in the empirical findings from existing work**
>
> Thank you for these references. We briefly summarize how these works relate to ours:
>
> - Zerroug et al. focus on visual reasoning and data-efficiency, combining representation learning and reasoning over them with repeatedly sampled (i.e. the train and test set has overlap), or procedurally-generated tasks (visual primitives aren't overlapping between training and testing). In contrast, our work centers on classification tasks that rely on strong vision representations and examines generalization when seen concept values at test time are combined in unseen ways.
> - Zhou et al. study data compositionality in NLP using primitive-level data augmentation to enhance generalization. While we find a similar conclusion that diverse training data is beneficial, our focus on the vision domain introduces challenges not encountered in language tasks (e.g. making primitive-level augmentations is simple in language, but not in vision; and scaling ID data quantity isn't sufficient to improve generalization).
> - Stone et al. modify architectures and losses with object masks to promote compositionality, which relies on injecting priors and using dataset-specific annotations. Our approach, however, investigates compositional generalization from a data-scaling perspective.
>
> We believe our work isolates a specific and underexplored aspect of compositional generalization in vision. Our work, to our knowledge, is the first to consider the scale of compositions across multiple dimensions: types of concepts, number of concept values, number of datapoints used in training, types of datasets, and densities of compositions in a controlled manner.

---

> > ### Comment · Reviewer_Nv93 · 2025-04-09
> >
> > I appreciate the authors for addressing my comments. However, my concern about real-world or complex scenarios extension has not been well addressed. Although the authors emphasized the controllability of the simple environment and highlighted that the current setting could be challenging enough for the evaluated models, I believe it is intuitive to investigate more complex and realistic data since the paper studied compositionality with a focus on data scaling. Moreover, it is still unclear whether the presented results could reflect or have implications for more challenging data and scenarios (the second weakness in my review). Based on this, I decided to keep my score.

---

> > > ### Author Response · Authors · 2025-04-09
> > >
> > > Thank you for your follow-up.
> > >
> > > We have additionally conducted experiments on the real-world dataset MPI3D [1], containing real photographs with controlled variation factors (examples in Figure 1). Full results are available at: https://anonymous.4open.science/r/7475686574/rebuttal_response.pdf. In short, all of our original claims hold: compositional generalisation remains challenging in from-scratch models (Figure 2), scaling ID data quantity alone does not help (Figure 3, top left), and increasing class diversity or combination coverage improves generalisation (Figure 3, top right and bottom). Pre-trained models show linear factorisation and near-perfect accuracy on colour (Figure 4, left), yet probing shows struggles with harder concepts like shape (Figure 4, right).
> > >
> > > While our environments are controlled, they allow us to systematically scale key data factors, and the consistent results across synthetic and real-world data suggest that the difficulties we observe reflect fundamental model limitations. We are not aware of any real-world datasets with known factor structures at the scale and level of control considered in this work. Without full control over the concepts we train on, or when using mislabeled or partially labeled real-world images, it would not be possible to correctly attribute which data factors drive successful generalisation.
> > >
> > > Finally, we note that Section 5 evaluates large-scale pre-trained models, trained on diverse real-world data, yet compositional generalisation remains hard--further supporting that this is a core challenge, not an artifact of the experimental setup and the type of data.
> > >
> > > In sum, we agree that real-world datasets are important for future work, but we believe this study provides an important first step toward understanding the key factors underlying compositional generalisation, understanding what feature space structure enables it, and to which extent current vision backbones support it.
> > >
> > > [1] Gondal, Muhammad Waleed, et al. "On the transfer of inductive bias from simulation to the real world: a new disentanglement dataset." Advances in Neural Information Processing Systems 32 (2019).

---

### Official Review · Reviewer_hh3A · 2025-03-13

**Overall Recommendation:** 3

**Summary:**

This paper examines whether data scaling improves compositional generalization in vision models, emphasizing the role of data diversity. Through controlled experiments with synthetic datasets, the authors show that models can achieve compositional generalization, but this ability depends on training data diversity rather than quantity.

**Claims And Evidence:**

Basically yes.

**Essential References Not Discussed:**

No

**Experimental Designs Or Analyses:**

Yes, please see "Methods And Evaluation Criteria"

**Methods And Evaluation Criteria:**

The experimental setup is somewhat unconvincing:
1) The authors report the average accuracy across all concepts at each epoch (Line 223). What is the significance of reporting average accuracy per epoch? Does it effectively reflect the model's final compositional generalization capability?
2) The use of two independent classification heads to predict each concept in a composition for evaluating compositional generalization is debatable (Line 230). Other setups, such as a similarity-based approach, should be considered to improve comprehensiveness of evaluation.
3) The authors claim to have evaluated three probe architectures (Line 255), but these are not mentioned in the experimental results. Additionally, the relevance of evaluating architectures with only minor differences (e.g., one hidden layer or activation function) remains debatable.
4) Does the authors isolate a single variable in their experiments to ensure a controlled comparison? For instance, Does the number of traing samples kept constant while increasing training data diversity? (Sec. 4.1)
5) Does the authors consider conducting experiments on real-world datasets?

**Other Comments Or Suggestions:**

N/A

**Other Strengths And Weaknesses:**

As mentioned above, this paper's topic has both theoretical and practical significance, but the experimental setup needs improvement.
Additionally, some parts of the writing require clarification. For example, on line 318, what does performance specifically refer to?

**Questions For Authors:**

See `Methods And Evaluation Criteria` and `Other comments Or Suggestions`.

**Relation To Broader Scientific Literature:**

This paper explores the impact of data on the compositional generalization ability of vision models, a topic of broad relevance. As deep learning increasingly depends on large-scale data pretraining, optimizing data collection and filtering with compositional generalization in mind can significantly enhance training efficiency and model performance.

**Theoretical Claims:**

Yes.

---

> ### Author Rebuttal · Authors · 2025-04-01
>
> Thank you for your thorough review. We will clarify the raised points in the revised manuscript.
>
> **L223: What is the significance of reporting average accuracy per epoch? Does it effectively reflect the model's final compositional generalization capability?**
>
> Thank you for pointing this out. We should have stated this more clearly. The metric reported in Line 223 corresponds to the best-performing epoch (i.e., oracle model selection), not an average over epochs. Specifically, we track the average accuracy across both concepts at each epoch and report the highest value observed during training. This provides an estimate of the *upper bound* on achievable performance under ideal model selection. We chose this evaluation to emphasize that even with perfect model selection, the performance gap between ID and OOD is present.
>
> **The use of two independent classification heads to predict each concept in a composition is debatable (Line 230)**
>
> If we understood your question correctly, by similarity-based, do you mean a CLIP-like objective? Our current design mirrors the supervision used in CLIP and standard classification settings, up to norm-rescaling of the features and weight vectors. The main difference is that our setup assumes a closed-vocabulary setting, and we believe that using two independent classifiers is appropriate since the concepts are independent.
>
> We would be happy to hear more about alternative formulations or suggestions on how to frame this classification problem!
>
> **The authors claim to have evaluated three probe architectures (Line 255), but these are not mentioned in the experimental results.**
>
> Our main goal was to evaluate the pre-trained models' representations as charitably as possible. We did evaluate multiple probe architectures, but our main text only reported the best-performing configuration.
>
> In response to this comment, we now include the average testing accuracies on both concepts across all datasets across 4 settings of data diversity (% of combinations in training), comparing linear probes and MLPs. MLPs most often performs better, but not universally across the datasets. The first number is the linear probe and the second is the MLP ([512, 512]):
>
> | Model | 25.0%| 50.0%| 75.0% | 93% |
> |----|---|---|----|----|
> | ResNet50 IN1K | 0.59 / 0.55 | 0.67 / 0.65 | 0.75 / 0.75 | 0.79 / 0.82 |
> | DINOv1 ResNet50 | 0.60 / 0.67 | 0.71 / 0.80 | 0.76 / 0.88 | 0.80 / 0.92 |
> | DINO ViT-L | 0.68 / 0.70 | 0.78 / 0.83 | 0.84 / 0.91 | 0.86 / 0.95 |
> | CLIP ViT-L | 0.61 / 0.64 | 0.70 / 0.74 | 0.75 / 0.79 | 0.76 / 0.84 |
>
> **Additionally, the relevance of evaluating architectures with only minor differences (e.g., one hidden layer or activation function) remains debatable.**
>
> While the architectural variations may appear small, we found that going beyond two hidden layers didn't result in substantial increase in performance. Below, we provide comparisons using the FSprites dataset (again for 4 diversity settings); we used these results as justification for focusing on simpler configurations in this work.
>
> | Model / Probe | 25%  | 50%  | 75%  | 93%  |
> |--------------------|------|------|------|------|
> | ResNet50 IN1K (Lin)| 0.38 | 0.52 | 0.57 | 0.66 |
> | ResNet50 IN1K (MLP)| 0.35 | 0.47 | 0.55 | 0.64 |
> | ResNet50 IN1K (Deep)|0.34 | 0.46 | 0.55 | 0.63 |
> | DINO R50 (Lin) | 0.40 | 0.56 | 0.64 | 0.74 |
> | DINO R50 (MLP) | 0.41 | 0.62 | 0.72 | 0.79 |
> | DINO R50 (Deep) | 0.41 | 0.60 | 0.71 | 0.79 |
> | DINO ViT-L (Lin) | 0.44 | 0.63 | 0.78 | 0.81 |
> | DINO ViT-L (MLP) | 0.46 | 0.65 | 0.81 | 0.90 |
> | DINO ViT-L (Deep) | 0.45 | 0.64 | 0.81 | 0.90 |
> | CLIP ViT-L (Lin)   | 0.41 | 0.56 | 0.68 | 0.74 |
> | CLIP ViT-L (MLP) | 0.43 | 0.60 | 0.73 | 0.85 |
> | CLIP ViT-L (Deep) | 0.42 | 0.59 | 0.72 | 0.82 |
> (Lin = linear probe, MLP = [512, 512], Deep = [512, 512, 512])
>
> **Does the number of training samples remain constant while increasing training data diversity? (Sec. 4.1)**
>
> Yes, we do control for this factor, although we acknowledge that it was not made sufficiently clear in the text. All experiments in Section 4.1 (except for Figure 4) use a fixed dataset size of 60,000 samples where available; for Shapes3D and CMNIST we used 30,000 samples.
>
> **Does the authors consider conducting experiments on real-world datasets?**
>
> We considered real-world datasets with known generative factors, but they are currently limited in compositional complexity. Moreover, real-world datasets offer sparse coverage of concept values, making controlled compositional generalization experiments intractable. We are open to suggestions for real-world datasets that offer controllability.
>
> Finally, although our datasets are synthetic, Section 5 evaluates pre-trained models used in real-world practice and tests how pre-trained vision models cope in compositional settings.
>
> **Line 318, what does performance specifically refer to?**
>
> By performance, we refer to the mean classification accuracy over the considered concept values.

---

> > ### Comment · Reviewer_hh3A · 2025-04-04
> >
> > The authors partly sovled my concerns. I think experiments sbould be more sufficient.

---

> > > ### Author Response · Authors · 2025-04-05
> > >
> > > Thank you for your follow-up. Could you please clarify which concerns you feel we have not addressed in our clarifications and additional results?
> > >
> > >
> > > **EDIT**:
> > > We have conducted additional experiments on a real-world dataset, MPI3D [1]. This dataset contains real photographs of various objects with controlled factors of variation, such as shape, colour, orientation, size, and position (examples shown in Figure 1). Please find the full results at the anonymised link:
> > > [https://anonymous.4open.science/r/7475686574/rebuttal_response.pdf](https://anonymous.4open.science/r/7475686574/rebuttal_response.pdf)
> > >
> > > In short, all of our original claims hold on this dataset as well. Specifically:
> > >
> > > - **From-scratch training**:
> > >   Compositional generalisation remains problematic. In Figure 2, we show that the model undergoes a large drop in performance for one of the concepts when evaluated on unseen combinations.
> > >
> > > - **Scaling ID data quantity**:
> > >   As shown in Figure 3 (top left), scaling ID data quantity alone does not help.
> > >   Instead, increasing either the number of target classes (top right) or the number of combinations (bottom) improves compositional generalisation performance.
> > >
> > > - **Pre-trained models**:
> > >   We again observe that linear factorisation is often present. In Figure 4 (left), we show that under this assumption, the models can classify the concepts well above random chance. In this case, all models show near-perfect accuracy on the colour concept.
> > >
> > > - **Probing with linear and non-linear classifiers**:
> > >   When probing the models (Figure 4, right), they continue to struggle, mostly on the shape concept.
> > >
> > > We would like to emphasise that these experiments represent the best we could achieve given the current constraints in compositional generalisation research and evaluation (particularly the availability of concept labels).
> > >
> > > We hope this additional evidence further clarifies the robustness of our findings across both synthetic and real-world data.
> > >
> > > [1] Gondal, Muhammad Waleed, et al. "On the transfer of inductive bias from simulation to the real world: a new disentanglement dataset." *Advances in Neural Information Processing Systems* 32 (2019).

---

### Official Review · Reviewer_FpUd · 2025-03-13

**Overall Recommendation:** 5

**Summary:**

Compositional generalization is a fundamental aspect of human intelligence, yet its emergence in machine learning, particularly in vision models, remains unclear. This paper investigates whether increasing data scale contributes to compositional generalization in vision models by systematically isolating the effects of dataset size, diversity, and compositional structure. Through behavioral evaluations and probing latent representations of both pretrained and from-scratch models, the authors find that data diversity—specifically, the ratio of exhaustive compositional samples—plays a crucial role in generalization, rather than sheer data quantity. Their findings suggest that compositional representations emerge only under conditions of sufficient data diversity, where representations become more structured and linearly organized in latent space, facilitating efficient generalization.

**Claims And Evidence:**

The claims presented in the paper are well-supported by experimental results and clearly articulated. The authors systematically control for dataset characteristics and demonstrate that compositional generalization is not merely a function of dataset size but rather of diversity in compositional coverage. Their analysis of pretrained models further substantiates their claims, revealing that large-scale vision models do not inherently develop compositional generalization unless trained on sufficiently diverse data.

**Essential References Not Discussed:**

A related study by Dorrell et al. (2024) suggests that modularity in latent representations arises from certain input distribution characteristics. This work may provide additional theoretical grounding for the claim that data diversity enhances compositional generalization by promoting structured latent spaces. Including a discussion on how their findings align or contrast with Dorrell et al.’s results would strengthen the paper’s positioning within the literature.
Reference: Dorrell, Will, et al. "Don't Cut Corners: Exact Conditions for Modularity in Biologically Inspired Representations." arXiv preprint arXiv:2410.06232 (2024).

**Experimental Designs Or Analyses:**

The experimental design is rigorous and well-structured, effectively isolating key variables affecting compositional generalization. The study employs controlled datasets where specific combinations of concepts are systematically included or excluded from training, allowing precise analysis of generalization behavior. The comparison between from-scratch models and large-scale pretrained models provides valuable insights into the role of pretraining. However, it would be beneficial to extend the analysis to other modalities, such as language models or vision-language models, to examine whether similar principles apply across domains.

**Methods And Evaluation Criteria:**

The methods employed in the study are appropriate for assessing compositional generalization in vision models. The authors introduce a systematic protocol that varies dataset size, concept values, and training combinations while keeping other factors controlled. Their use of synthetic datasets allows precise manipulation of compositional factors, ensuring that their results are not confounded by spurious correlations. The evaluation criteria, including out-of-distribution (OOD) accuracy and probing latent representations, align well with the problem at hand and provide clear insights into model generalization.

**Other Comments Or Suggestions:**

1. Clarify Definitions: The paper would benefit from a clearer distinction between "features" and "concepts." Are features referring to learned latent representations, while concepts are explicit or implicit properties of the stimuli?
2. Further Exploration of Takeaway 4.1: Consider investigating non-uniform distributions of n−k concept combinations to determine whether concepts that exhibit lower degradation in uniform settings maintain robustness under more naturalistic data distributions.
3. Additional Experimentation: If space allows, evaluating whether the results extend to language models or vision-language models would strengthen the paper's impact.

**Other Strengths And Weaknesses:**

The paper is well-written and clearly structured, making it an enjoyable read. The authors conduct an extensive set of experiments that provide strong empirical support for their claims. A notable strength is their systematic approach to controlling dataset characteristics, which allows them to make precise claims about the role of data diversity. However, one potential limitation is the exclusive focus on vision models—extending the study to multimodal or language models could provide a more comprehensive understanding of compositional generalization. Additionally, it would be interesting to explore whether the findings hold for different architectural choices, such as transformer-based vision models.

**Questions For Authors:**

1. **Compositionality and Simplicity Bias**
   - In Section 2, why does the simplicity bias lead to model preference for spurious correlations? And why is this issue more pronounced for underrepresented concepts?

2. **Dataset Combinatorial Calculations**
   - In Section 3.1, when calculating the number of possible unique images, should the formula be \( n \times k + k \times (n-k) \) instead?
   - Since we consider \( n \times k \) for the first dimension while the remaining \( n-k \) concept values for the second dimension should also be combined with possible \( k \) values from the first dimension.

3. **Figure 3 Analysis**
   - What is the boundary value of \( k \) for achieving the observed compositional generalization improvements?
   - Additionally, for fixed maximum target classes, does the improvement hold for lower \( n \) values?

4. **Figure 4 Clarification**
   - What is the y-axis measuring—OOD accuracy or drop in performance?
   - Also, why does FSprites exhibit lower values in general? What factors make it more challenging compared to other datasets?

5. **Figure 7 Interpretation**
   - Does the gap indicate that the retrained model demonstrates better compositional generalization performance than a from-scratch model, or does it reflect the limitations of the pretrained model?

6. **Linear Probing in Section 5**
   - What is the \( k \) value used in the linear probing analysis?
   - How does it compare to the threshold required for compositional representations to emerge?

**Relation To Broader Scientific Literature:**

The paper is highly relevant to researchers studying generalization in machine learning, particularly in vision and compositionality. It also holds significance for neuroscience and cognitive science communities, as it provides insights into how structured learning environments influence compositional learning. The findings can inform experimental design in cognitive science by emphasizing the importance of exhaustive combinatorial sampling. Additionally, the study is valuable for researchers working on dataset construction and training strategies, as it suggests guidelines for optimizing training data diversity to improve generalization.

**Theoretical Claims:**

The paper does not focus on proving new theoretical results but instead builds on existing theoretical insights into compositional learning. The experimental findings align with prior work on neural network biases and structure in latent space, supporting claims about the importance of data diversity for compositional generalization. The authors also introduce a theoretical analysis showing that only a small number of compositional samples are required for perfect generalization if the feature space is structured compositionally, which is empirically validated in their experiments.

---

> ### Author Rebuttal · Authors · 2025-04-01
>
> Thank you for your thorough review. We will add clarifications to the updated manuscript.
>
> **Clarify of definitions**
>
> In our terminology, “concepts” refer to interpretable properties of the data (e.g., colour, shape, size), while “features” refer to the latent representations extracted by the model.
>
> **Investigating non-uniform distributions**
>
> The goal in our setup was to ensure uniform coverage over observed combinations to avoid introducing imbalances across concept values. While certainly interesting, we are careful in complexifying the data setting in a way that makes the problem harder.
>
> **Evaluating language models and VLM.**
>
> Extending to language or vision-language models is challenging. Autoregressive models lack comparable representations. For non-autoregressive ones like CLIP, we probed the vision encoder; probing the text encoder is harder, as it requires generating many prompts to describe concept values. Nonetheless, we consider this an important direction for future work.
>
> **Transformer-based vision models.**
>
> Due to space constraints in this response, we couldn’t include the results for the vision transformer here. Another reviewer also expressed interest in the ViT’s performance--please refer to the table in the response to reviewer `TQRD`.
>
> **Related study by Dorrell et al. (2024).**
>
> Thank you for pointing us to this excellent reference--we will be sure to discuss it in more detail. In short, this work, as well as the work it builds on (Whittington et al., 2023), characterises when the minimal-energy non-negative solution is modular, i.e when each neuron responds to a single task factor. We believe both this framework and ours can be viewed in a similar light. The key distinction lies in the goal: while their focus is on the structure of solutions over the observed data, our work studies whether representations can be predicted from their constituent concept values under novel combinations in the test set, and how data factors influence the structure of representations.
>
> **Simplicity bias and spurious correlations, underrepresented concepts**
>
> Our reasoning is based on prior observations in the debiasing literature. When training data exhibits underspecification--i.e., when multiple hypotheses are consistent with the data--a model may favour simpler hypotheses. In such cases, it may learn to generalize to majority groups and memorize the underrepresented groups [a]. In compositional generalization settings, this happens when some combinations are missing entirely (as in OOD splits), and models default to entangled solutions instead of learning disentangling concepts in their representation. Concepts, especially in the real world, are highly co-varying in some combinations and not in others, even if the observed concept pairs are not entangled. Because of this sampling bias, models may exploit such spurious correlations.
>
> [a] Sagawa, Shiori, et al. "An investigation of why overparameterization exacerbates spurious correlations." ICLR 2020.
>
> **In Section 3.1, formula for unique images**
>
> In Section 3.1, we were referring specifically to the number of training combinations observed in the $(n,k)$ framework, but we will clarify this distinction in the revised text.
>
> **Figure 3: boundary value of $k$**
>
> If we understood your question correctly: The boundary value of $k$ corresponds to $n-1$, that is, training on all but one combination per concept value.
>
> **Figure 3: For maximum target classes, does the improvement hold for lower $n$?**
>
> Yes, and we believe Figure 10 in the appendix demonstrates this: for fixed $k$, increasing $n$ lowers generalization performance; conversely, for fixed $n$, increasing $k$ improves it.
>
> **Figure 4: What is the $y$-axis measuring? Also, why does FSprites exhibit lower values in general?**
>
> The $y$-axis in Figure 4 shows OOD accuracy. FSprites performs worse than other datasets, likely due to its more complex and visually similar shapes, which are harder to disentangle. Also, orientation may not align well with shape features, encouraging memorisation of joint shape-orientation co-variations instead of separate concept learning.
>
> **Figure 7: Does the gap indicate that the retrained model demonstrates better compositional generalization performance than a from-scratch model, or does it reflect the limitations of the pretrained model?**
>
> It indicates both. The retrained model benefits from better compositional generalization due to exposure to more diverse or better-structured training data. However, the presence of a gap also reflects the limitations of the pretrained models to generalize compositionally.
>
> **Linear Probing in Section 5: What is the $k$ value used in the linear probing analysis? How does it compare to the threshold required for compositional representations to emerge?**
>
> We varied $k$ from 2 to $n-1$, showing only aggregated results across datasets. While the exact threshold varies, $k=2$ is theoretically the minimum needed per dataset.

---

### Decision · Program_Chairs · 2025-05-01

[review text omitted: it was posted to a different submission]